EMBO
Molecular Medicine

# Loss of the mitochondrial *i*-AAA protease YME1L leads to ocular dysfunction and spinal axonopathy

Hans-Georg Sprenger[1,2], Gulzar Wani[2], Annika Hesseling[2], Tim König[2,†], Maria Patron[1,2], Thomas MacVicar[1,2], Sofia Ahola[1,2], Timothy Wai[2,‡,§], Esther Barth[2], Elena I Rugarli[2], Matteo Bergami[2,3] & Thomas Langer[1,2,3,*] (iD)

## Abstract

Disturbances in the morphology and function of mitochondria cause neurological diseases, which can affect the central and peripheral nervous system. The *i*-AAA protease YME1L ensures mitochondrial proteostasis and regulates mitochondrial dynamics by processing of the dynamin-like GTPase OPA1. Mutations in *YME1L* cause a multi-systemic mitochondriopathy associated with neurological dysfunction and mitochondrial fragmentation but pathogenic mechanisms remained enigmatic. Here, we report on striking cell-type-specific defects in mice lacking YME1L in the nervous system. YME1L-deficient mice manifest ocular dysfunction with microphthalmia and cataracts and develop deficiencies in locomotor activity due to specific degeneration of spinal cord axons, which relay proprioceptive signals from the hind limbs to the cerebellum. Mitochondrial fragmentation occurs throughout the nervous system and does not correlate with the degenerative phenotype. Deletion of *Oma1* restores tubular mitochondria but deteriorates axonal degeneration in the absence of YME1L, demonstrating that impaired mitochondrial proteostasis rather than mitochondrial fragmentation causes the observed neurological defects.

**Keywords** axonal degeneration; microphthalmia; mitochondrial proteostasis; OMA1; YME1L

**Subject Categories** Genetics, Gene Therapy & Genetic Disease; Neuroscience

See also: **ZMA Chrzanowska-Lightowlers & RN Lightowlers** (January 2019)

## Introduction

Many prevalent and rare neurodegenerative disorders have been associated with mitochondrial deficiencies, which affect central as well as peripheral parts of the nervous system (Nunnari & Suomalainen, 2012; Kawamata & Manfredi, 2017; Viscomi & Zeviani, 2017; Nissanka & Moraes, 2018). Mitochondria are central metabolic organelles and therefore of pivotal importance for neuronal survival. They provide most neuronal ATP by oxidative phosphorylation (OXPHOS) and thus ensure axonal trafficking of organelles and synaptic transmission. Impaired respiration causes a neuronal energy crisis, inhibits mitochondrial motility, and leads to axonal degeneration in disease, often in a highly neuron-specific manner (Kawamata & Manfredi, 2017; Misgeld & Schwarz, 2017; Viscomi & Zeviani, 2017; Nissanka & Moraes, 2018). Over half of all patients suffering from a mitochondrial disease for instance develop ocular complications, which in some cases are combined with brain atrophy or specific degeneration of motor neurons in the spinal cord (Yu-Wai-Man *et al*, 2016). The striking cell-type specificity of mitochondrial disorders affecting the nervous system is poorly understood.

Similar to OXPHOS defects, deficiencies in other metabolic functions of mitochondria can cause neurodegenerative disorders, as can disturbances in the morphology of mitochondria (Delettre *et al*, 2000; Chen *et al*, 2007; Gerber *et al*, 2017), which is intimately coupled to mitochondrial function and quality control (Youle & van der Bliek, 2012). Dynamin-like GTPases acting on the mitochondrial outer (OM) and inner membrane (IM) ensure the balanced fusion and fission of mitochondria and determine their structure (Friedman & Nunnari, 2014; Mishra & Chan, 2014). Mutations in these GTPases cause complex neurological disorders including dominant optic atrophy (DOA) and Charcot–Marie–Tooth syndrome type 2A (Delettre *et al*, 2000; Zuchner *et al*, 2006; Waterham *et al*, 2007; Gerber *et al*, 2017). At the level of the IM, the GTPase optic atrophy 1 (OPA1) mediates mitochondrial fusion and regulates cristae morphogenesis (Cogliati *et al*, 2016). Loss of OPA1 inhibits mitochondrial fusion and results in the fragmentation of the mitochondrial network, which is linked to axonal transport defects and mitophagy (Twig *et al*, 2008; Gomes *et al*, 2011). OPA1 undergoes proteolytic processing leading to the balanced accumulation of short (S-) and

1  Max-Planck-Institute for Biology of Ageing, Cologne, Germany
2  Institute of Genetics and Cologne Excellence Cluster on Cellular Stress Responses in Aging-Associated Diseases (CECAD), University of Cologne, Cologne, Germany
3  Center for Molecular Medicine, University of Cologne, Cologne, Germany
  *Corresponding author. Tel: +49 221 37 970 500; E-mail: langer@age.mpg.de
  †Present address: Department of Neurology and Neurosurgery, Montreal Neurological Institute, McGill University, Montreal, QC, Canada
  ‡Present address: Institut Pasteur, CNRS UMR 3691, Mitochondrial Biology Group, Paris, France
  §Present address: Sorbonne Paris Cité, Paris Descartes University, Paris, France

 

membrane-anchored long forms (L-) of OPA1 (MacVicar & Langer, 2016). Whereas either L-OPA1 or S-OPA1 is sufficient to preserve cristae structure and respiration (Anand et al, 2014; Del Dotto et al, 2017; Lee et al, 2017), L-OPA1 is required to mediate mitochondrial fusion (Tondera et al, 2009; Ban et al, 2017; Chen & Chan, 2017). Two IM proteases can cleave L-OPA1 at neighboring sites and limit fusion: the stress-activated peptidase OMA1 (Ehses et al, 2009; Head et al, 2009) and the ATP-dependent i-AAA protease YME1L (Griparic et al, 2007; Song et al, 2007). The processing of OPA1 by these two proteases offers additional possibilities of regulation allowing to adapt mitochondrial morphology to distinct physiological cues (Mishra et al, 2014; MacVicar & Langer, 2016).

OMA1-deficient mice show impaired OPA1 cleavage and exhibit marked metabolic alterations, highlighting the importance of mitochondrial dynamics for metabolic control (Quiros et al, 2012). The loss of OMA1 impairs the expression of lipid and glucose metabolic enzymes in adipocyte tissues, which results in defective thermoregulation, reduced energy expenditure, and obesity (Quiros et al, 2012). However, the function of OMA1 in the nervous system remains largely elusive. OMA1 activation and mitochondrial fragmentation occurs under various stress conditions, such as mitochondrial depolarization, heat and oxidative stress, and occurs in apoptotic cells (Baker et al, 2014; Jiang et al, 2014; Zhang et al, 2014). Accordingly, OMA1 deficiency reduces the sensitivity to apoptosis (Quiros et al, 2012; Jiang et al, 2014; Korwitz et al, 2016). On the other hand, cells lacking the second OPA1 cleaving peptidase YME1L show an increased vulnerability to apoptosis due to OMA1 activation under these conditions (Stiburek et al, 2012; Anand et al, 2014).

YME1L preserves mitochondrial proteostasis and function acting both as a quality control and as a regulatory enzyme in the IM (Quiros et al, 2015; Levytskyy et al, 2017). Besides cleaving OPA1, it degrades damaged or non-assembled IM proteins such as respiratory chain subunits or TIMM17A, a subunit of a protein translocase in the IM (Stiburek et al, 2012; Rainbolt et al, 2013). Moreover, YME1L mediates the turnover of the short-lived lipid transfer proteins PRELID1 and STARD7 in the intermembrane space (IMS) and therefore also regulates mitochondrial phospholipid homeostasis (Potting et al, 2013; Saita et al, 2018). YME1L is essential for embryonic development in mice, whereas YME1L deficiency in adult cardiomyocytes causes dilated cardiomyopathy and heart failure (Wai et al, 2015). The loss of YME1L results in OMA1 activation and mitochondrial fragmentation (Anand et al, 2014; Wai et al, 2015). Cardiac function was maintained upon ablation of Oma1 in

these mice, which leads to the accumulation of L-OPA1 and the formation of tubular mitochondria. Thus, imbalanced mitochondrial dynamics is deleterious in the heart and OMA1 promotes cardiomyocyte cell death.

Homozygous recessive mutations in human YME1L, which destabilize YME1L and trigger mitochondrial fragmentation, cause a neuromuscular disorder with intellectual disability, motor developmental delay, optic atrophy as well as ataxia and movement deficiencies (Hartmann et al, 2016). However, how YME1L deficiency affects neuronal function remained enigmatic. Here, we show that the loss of YME1L in the nervous system impairs mitochondrial morphology and proteostasis throughout the nervous system but results in striking cell-type-specific neurological defects in mice: While newborn mice show microphthalmia with retinal inflammation and cataracts, spinal cord axons of the dorso-lateral column progressively degenerate with age, impairing coordinated movements. Axonal degeneration correlates with mitochondrial transport defects in YME1L-deficient, cultured neurons. Additional deletion of Oma1 restored mitochondrial morphology in vivo but deteriorated axonal degeneration, suggesting that impaired mitochondrial proteostasis rather than mitochondrial fragmentation causes mitochondrial trafficking defects and axonal loss.

## Results

### Loss of YME1L in the nervous system causes microphthalmia, cataracts, and retinal inflammation

We mated Yme1l[fl/fl] mice (Wai et al, 2015) with mice expressing the Cre recombinase under the control of the nestin promoter, which is active specifically in neuronal and glial cell precursors (Nestin-Cre) (Tronche et al, 1999). Homozygous offsprings (NYKO mice for Nestin-Yme1l-Knock Out mice) that lack YME1L in the nervous system were born at normal Mendelian ratios. However, newborn NYKO mice showed dramatic microphthalmia and developed cataracts (Fig 1A). The nestin promoter is activated at embryonic day 11 in the nervous system and drives Cre expression in retinal progenitor cells in the optic cup, which differentiate to various retinal cell types starting from embryonic day 12 (MacPherson et al, 2004; Adler & Canto-Soler, 2007). Thus, NYKO mice lack YME1L in all retinal cell types. A histological analysis revealed an overall normal organization of the retina with unaltered thickness of different retinal layers and no dramatic reduction of nuclei in the outer and

**Figure 1.  Loss of YME1L in the nervous system causes microphthalmia, cataracts, and retinal inflammation.**

A   Representative images of eyes and lenses from 6- to 7-week-old wild-type (WT) and nervous system-specific YME1L knockout (NYKO) mice. Orange dashed lines mark eye morphology. Scale bars, 5 mm.

B   Retinal sagittal cross sections from 6- to 7-week-old mice stained with hematoxylin and eosin. NFL = nerve fiber layer, IPL = inner plexiform layer, INL = inner nuclear layer, OPL = outer plexiform layer, ONL = outer nuclear layer, R&C = rods and cones. Scale bars, 30 μm.

C   Quantification of nuclei in OPL (area = 1,000 μm²) from retina cross sections of 6- to 7-week-old WT (n = 4) and NYKO (n = 4) mice.

D   Immunoblot analysis of retinal lysates from 6- to 7-week-old WT and NYKO mice. GFAP was used as a marker for reactive astrogliosis and SDHA as a loading control.

E   mRNA levels of proinflammatory cytokines and NF-κB target genes from 6- to 7-week-old retinas (WT, n = 5; NYKO, n = 5). Transcript levels were normalized to Hprt mRNA levels.

F   mRNA levels of Fgf21 from 6- to 7-week-old retinas (WT, n = 5; NYKO, n = 5). Transcript levels were normalized to Hprt mRNA levels.

G   Transmission electron micrographs of optic nerves from 6- to 7-week-old WT and NYKO mice. Scale bars, 2 μm.

Data information: Data were analyzed using unpaired t-test, *P ≤ 0.05, **P ≤ 0.01, ***P ≤ 0.001, ****P ≤ 0.0001, ns = not significant. Data are means ± SEM.
Source data are available online for this figure.

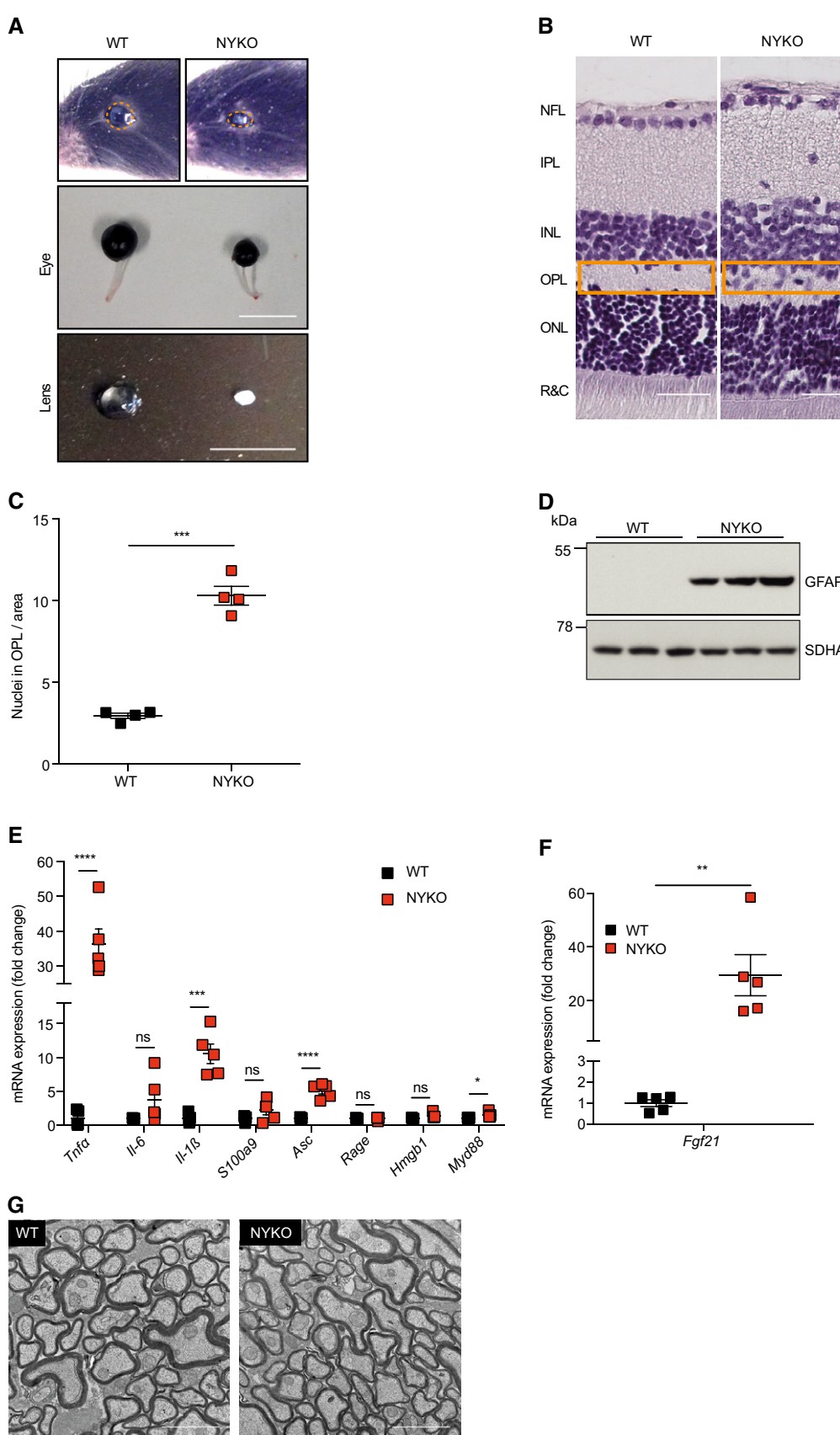

**Figure 1.**

inner nuclear layer (ONL, INL; Fig 1B, Appendix Fig S1A). The outer plexiform layer (OPL), however, appeared disorganized and contained nuclei that were absent in the OPL of wild-type retinas (Fig 1B and C). Moreover, the accumulation of the glial fibrillary acidic protein (GFAP) indicated neuroinflammation (Fig 1D), which was further substantiated by the increased expression of the proinflammatory cytokines and NF-κB target genes *Tnfα, Il-1β, Asc, and Myd88* in retinas of 6-week-old NYKO mice (Fig 1E). These findings are reminiscent of inflammatory responses observed in muscular OPA1 deficiency (Rodriguez-Nuevo *et al*, 2018). Similar to this model, we observed increased expression of *Fgf21* (Fig 1F). However, mtDNA levels remained unaffected in NYKO mice contrasting muscular OPA1 deficiency (Appendix Fig S1B).

We conclude that the loss of YME1L in retinal progenitor cells triggers microphthalmia with cataracts and retinal inflammation. Notably, the analysis of optic nerves in 6-week-old NYKO mice by transmission electron microscopy (TEM) did not reveal axonal degeneration or myelination defects, suggesting that retinal ganglion cells are not broadly affected (Fig 1G and Appendix Fig S1G). Previously identified mitochondrial proteins associated with the development of microphthalmia, namely HCCS and COX7B, were shown to induce a reactive oxygen species (ROS)-dependent induction of caspase 9-mediated cell death (Indrieri *et al*, 2012, 2013). However, neither the levels of mRNA expression of *Hccs* or *Cox7b*, nor the amount of cytochrome c or ROS scavenger enzymes were affected in retinas of NYKO mice (Appendix Fig S1C–E). Moreover, immunoblot analysis of caspase 8 and 9 revealed no activation in retinas deleted for YME1L (Appendix Fig S1F), indicating that the loss of YME1L induces microphthalmia along a different pathway.

## YME1L ensures locomotor activity and coordinated movements

With increasing age, NYKO mice gained less body weight when compared to heterozygous littermates or control mice (Fig 2A). Nuclear magnetic resonance tomography of 31-week-old mice revealed an overall reduced fat mass, with general and subcutaneous white adipose tissue being similarly affected (Figs 2B and EV1A). The whole-body energy expenditure of 26-week-old NYKO mice was significantly increased during the day and night period (Figs 2C and EV1B), although the activity levels of NYKO and control mice were similar (Fig EV1C). The impaired ability of NYKO mice to use the food containers hung at the sensor in the cages prevented a reliable determination of the food intake in these experiments and pointed to locomotor deficiencies of NYKO mice.

We therefore performed walking beam tests and monitored the time needed for the mice to traverse the beam (Fig 2D). NYKO mice were phenotypically normal at 6 weeks but showed a progressive locomotor impairment starting at 17 weeks of age (Fig 2D). Similarly, we observed an increasing number of slips per run, which was pronounced in 31-week-old NYKO mice shortly before we had to sacrifice the animals for ethical reasons. These locomotor deficiencies were absent in heterozygous animals excluding deleterious effects of the presence of the Cre recombinase (Fig 2D and E). Remarkably, this progressive degenerative phenotype was restricted to the hind limbs, which were positioned abnormally (Fig 2F, Movies EV1 and EV2). In agreement with these observations, grip

strength tests revealed that NYKO mice but not heterozygous littermates developed a dramatic impairment in their grip strength at 31 weeks, with their hind limbs being primarily affected (Fig EV1D). We therefore conclude that the absence of YME1L in the mouse nervous system causes a late-onset progressive locomotor impairment of the hind limbs.

## Neuroinflammation and axonal degeneration in spinal cord of NYKO mice

Locomotor control depends on the integration of neurons in the brain. An impaired locomotor activity correlates with brain atrophy and inflammation in many mitochondrial diseases (Maltecca *et al*, 2009; Almajan *et al*, 2012; Johnson *et al*, 2013; Ignatenko *et al*, 2018). We therefore analyzed brain morphology and brain weight in NYKO mice. Unexpectedly, brains of NYKO mice were morphologically normal and showed no signs of atrophy up to 31 weeks of age (Fig 3A). Nissl and Calbindin stainings of different brain areas of NYKO mice were indistinguishable from those of control mice (Fig 3B and C and Appendix Fig S2A). Moreover, loss of YME1L in the brain did not lead to neuroinflammation. We did not observe the accumulation of neuroinflammatory markers, such as GFAP and ionized calcium binding adaptor molecule 1 (IBA1), upon immunohistochemical and immunoblot analysis of brain tissue from 31-week-old NYKO mice (Fig 3D and Appendix Fig S2A). The pro-inflammatory cytokines *Tnfα, Il-6,* and *Il-1β* were expressed at similar levels in NYKO and control mice (Appendix Fig S2B). These results exclude general brain atrophy as the cause for the impaired locomotor activity of aged NYKO mice.

Coordinated movement also relies on the integrity of neurons in the spinal cord (Fig 4A). We therefore examined spinal cords of NYKO mice in further experiments. Transverse spinal cord sections were stained with toluidine blue, which allows detection of myelin dense bodies (MDBs). MDBs are a hallmark of axonal degeneration, as they accumulate when the myelin sheath collapses into the area formerly occupied by the axon (Jeong *et al*, 2011). Non-symptomatic 6-week-old NYKO mice did not show any morphological abnormalities in the white matter of the spinal cord (Fig 4B and C). MDBs were not detected in the different tracts of the spinal cord. Moreover, we did not observe changes in myelination by TEM (Fig EV2A and B). However, MDBs accumulated in the dorso-lateral tracts of the spinal cord of 17-week-old NYKO mice (Fig 4B and C). They formed more prominently in the cervical part of the spinal cord than in the lumbar part, whereas in aged animals, they were present to the same extent in both parts (Fig 4B and C). Strikingly, axons of the dorso-lateral tract were predominantly affected by the loss of YME1L. We observed only small numbers of MDBs in other ventral and ventro-lateral tracts of aged NYKO mice (Fig EV2C and D). In agreement with the degeneration of axons in the spinal cord, we observed upregulation of GFAP and increased expression of proinflammatory cytokines in the spinal cord of 31-week-old NYKO mice, indicating neuroinflammation (Fig 4D and E).

We conclude from these experiments that loss of YME1L leads to axonal degeneration specifically in the dorso-lateral tracts of the spinal cord. These tracts are the main somatosensory pathways communicating with the cerebellum and harbor long ascending

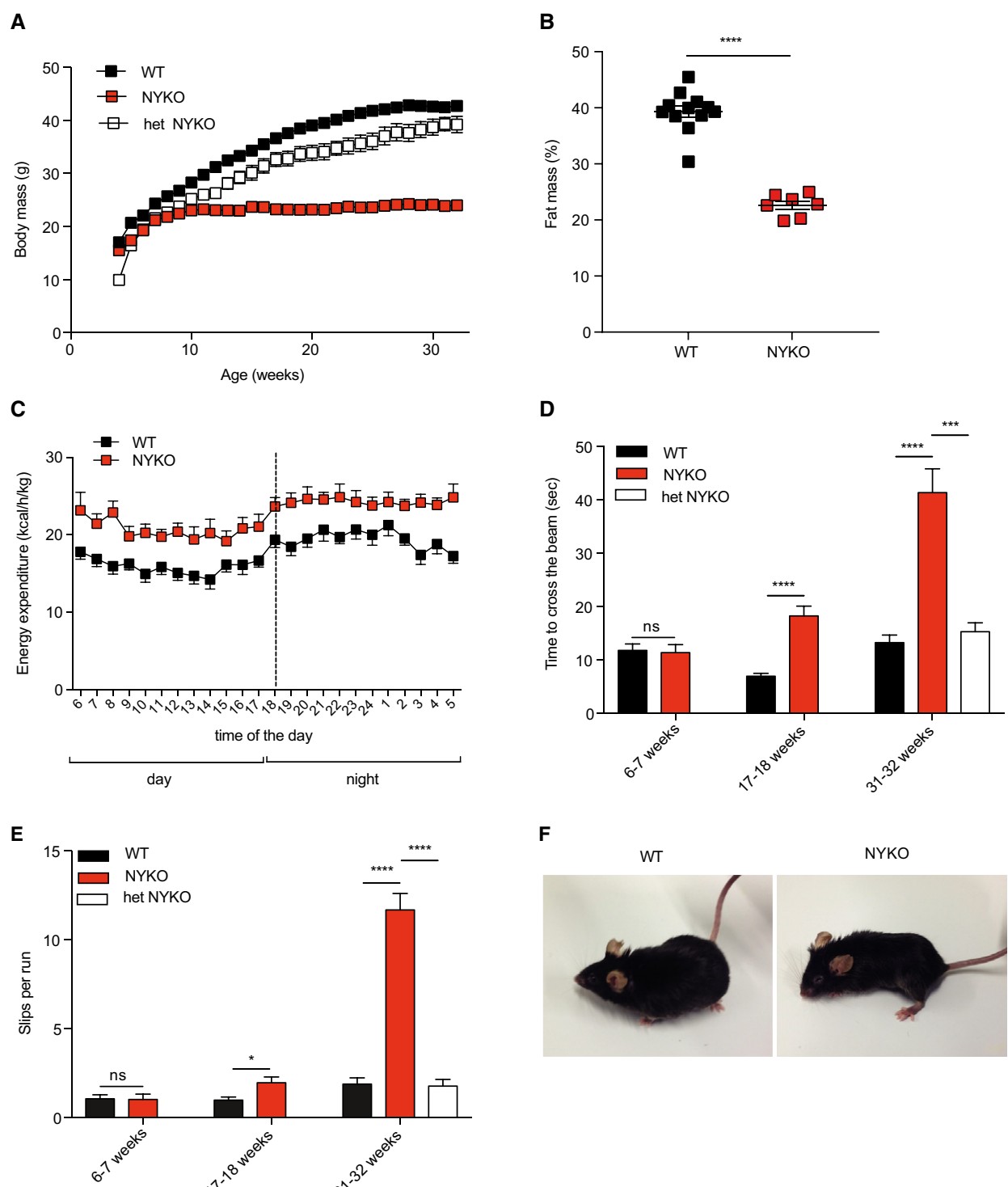

**Figure 2.  NYKO mice manifest locomotor impairment of hind limbs.**

A      Mean body weight of male wild-type (WT, *n* = 15), NYKO (*n* = 12), and heterozygous NYKO (het NYKO, *n* = 7) mice from 4 to 32 weeks.

B      Body composition analysis by nuclear magnetic resonance (NMR) of 31- to 32-week-old male WT (*n* = 12) and NYKO (*n* = 7) mice.

C      Whole-body energy expenditure per kg lean mass of 26-week-old male WT (*n* = 5) and NYKO (*n* = 5) mice after disease onset.

D, E   Walking beam test of 6- to 7-week-old (WT, *n* = 10; NYKO, *n* = 10), 17-18-week-old (WT, *n* = 8; NYKO, *n* = 9), and 31- to 32-week-old mice (WT, *n* = 12; NYKO, *n* = 19; het NYKO, *n* = 9). Hind limb slips per run are shown in (E).

F      Representative images of 31- to 32-week-old WT and NYKO mice showing aberrant positioning of hind limbs.

Data information: Unpaired *t*-test was used for the comparison of two groups, ordinary one-way ANOVA for the comparison of three groups. *$P \leq 0.05$, *** $P \leq 0.001$, ****$P \leq 0.0001$. Data are means ± SEM.

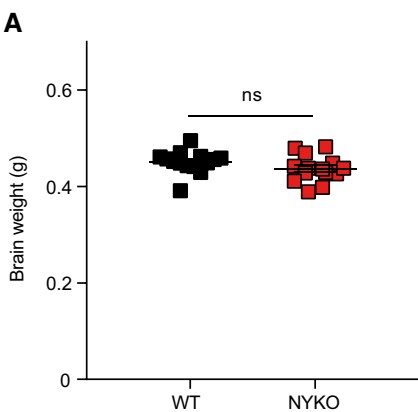

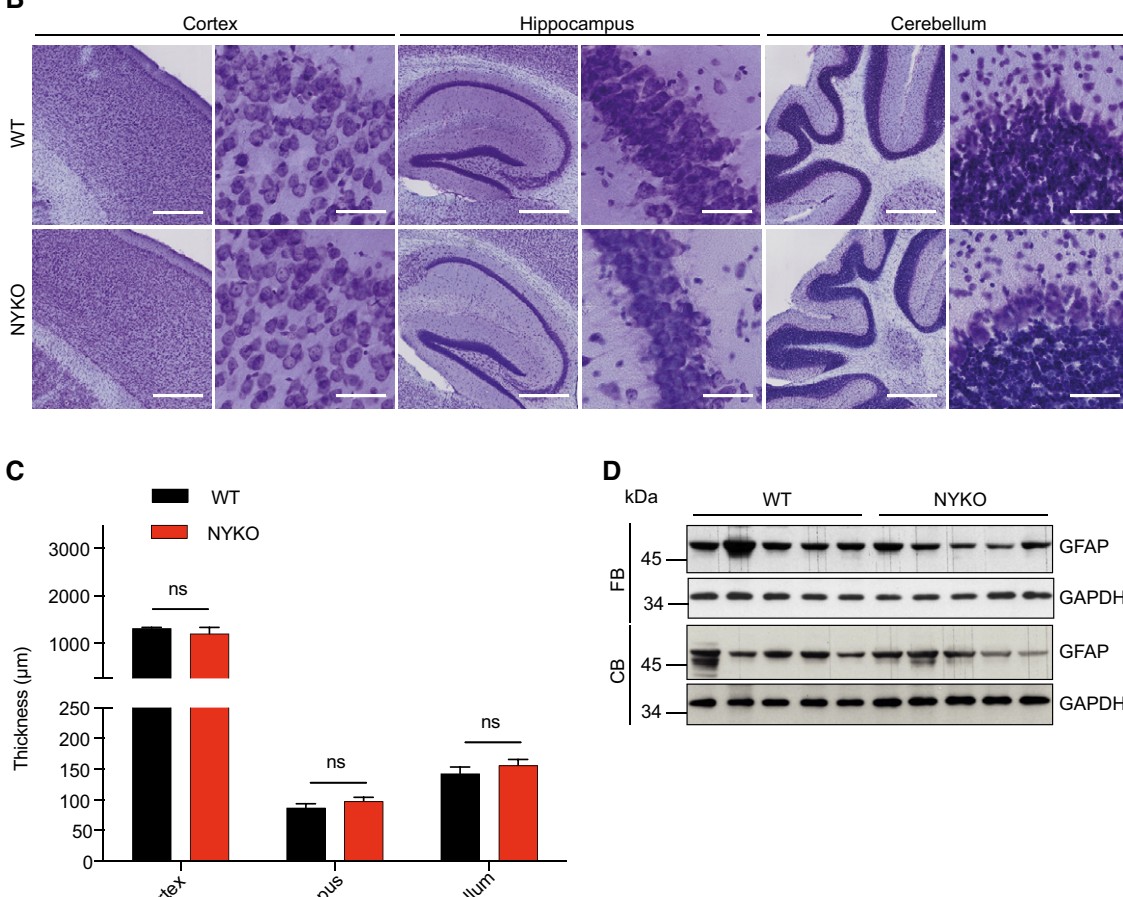

**Figure 3. The loss of YME1L in the nervous system does not trigger brain atrophy.**

A   Brain weights were monitored at 31–32 weeks of age (WT, *n* = 15; NYKO, *n* = 14).

B   Nissl stainings of sagittal sections across brain regions from 31- to 32-week-old WT and NYKO mice. 4× and 40× enlargements are shown in the left and right panels, respectively. Scale bars, 4× = 500 μm, 40× = 50 μm.

C   Morphometric analysis of different brain regions from 31- to 32-week-old WT (*n* = 6) and NYKO (*n* = 6) mice. GCL = Granule cell layer, ML = molecular layer.

D   Immunoblot analysis of tissue lysates (FB = forebrain; CB = cerebellum) from 31- to 32-week-old WT and NYKO mice, using GFAP-specific and, as a loading control, GAPDH-specific antibodies.

Data information: Unpaired *t*-test, ns = not significant. Data are means ± SEM.
Source data are available online for this figure.

neurons responsible for the relay of proprioceptive sensory signaling from the hind limbs (Hantman & Jessell, 2010). Therefore, axonal degeneration in these tracts likely explains locomotor deficiencies that become apparent in NYKO mice at 17 weeks of age, that is, concomitantly with the accumulation of MDBs in dorso-lateral tracts of the spinal cord.

### YME1L is required for efficient axonal transport of mitochondria

The accumulation of MDBs in cervical regions prior to lumbar regions in parts of the spinal cord that harbor ascending axons suggests dying-back degeneration in the absence of YME1L. Axonal maintenance depends on the efficient transport of mitochondria to ensure the supply with ATP and other metabolites (Sheng, 2017). We therefore examined whether mitochondrial transport defects may precede axonal degeneration in NYKO mice. To this end, neurons were isolated from cortices of $Yme1l^{fl/fl}$ mice and examined by live imaging after expression of plasmids encoding Cre-GFP or control GFP alone (Fig EV3A). Cre-mediated recombination of floxed $Yme1l$ alleles was confirmed by PCR (Appendix Fig S3) and microscopic analysis confirmed that Cre-expressing neurons display a mitochondrial network characterized by fragmented and clumped mitochondria (Fig 5A). No overt alterations in terms of morphology or signs of neurite degeneration were observed in cultured neurons lacking YME1L (Figs 5A and EV3). Morphometric analysis of neurons at 7 days $in\ vitro$ (DIV) revealed minor changes in the total length of the dendritic tree, whereas the number of primary dendrites as well as the overall axonal arbor appeared largely similar between control and $Yme1l$-deficient neurons (Figs 5B–E and EV3A). By DIV7, analysis of axonal trafficking in control neurons revealed that a substantial fraction of all mitochondria (over 60%) was stationary, leaving only about 35% of them being motile during 2 min of imaging (Fig 5 F and G). Interestingly, we observed a striking defect in the fraction of mitochondria specifically subjected to anterograde but not retrograde transport in neurons deleted for YME1L (Fig 5 F and G). Consistent with these findings, by DIV 10–14 the overall proportion of moving mitochondria in $Yme1l$-deficient neurons appeared significantly reduced compared to control neurons (21.2% in $Yme1l^{-/-}$ versus 38.8% in controls) (Fig 5H and I, Movies EV3 and EV4). In agreement with a defect in mitochondrial transport in $Yme1l$-deficient neurons, the axonal area occupied by mitochondria was reduced 2.5-fold when compared to control neurons (Fig 5J). Interestingly, the observed impairment

in mitochondrial trafficking due to the absence of YME1L did not prevent synapse formation during the investigated time window. By DIV12, analysis of Cre-GFP and GFP-only electroporated neurons revealed that the expression of vGlut1 (vesicular glutamate transporter 1) and vGat (vesicular GABA transporter), which are classic markers associated with excitatory and inhibitory synapses, was virtually undistinguishable between experimental groups (Fig EV3B and C).

Collectively, lack of YME1L is sufficient to impair anterograde axonal transport of mitochondria in cultured neurons.

### Late-onset defects in respiration and cristae structure in YME1L-deficient spinal cord mitochondria

Since impaired respiration and ATP production may account for mitochondrial transport defects in the absence of YME1L, we determined respiratory activities in YME1L-deficient spinal cord mitochondria. Substrate-driven respiration in the phosphorylating and uncoupled state was unaffected in mitochondria isolated from spinal cords of 6-week-old NYKO mice, while we observed a mild respiratory dysfunction in aged, 31-week-old NYKO spinal cord mitochondria (Figs 6A and B). Consistently, steady-state levels of selected OXPHOS subunits and the assembly of OXPHOS complexes were not affected in mitochondria that were isolated from spinal cords of 6-week-old NYKO mice but accumulated at reduced levels in spinal cord mitochondria of aged NYKO mice (Appendix Fig S4A–D).

The assembly of respiratory supercomplexes and respiratory efficiency has been linked to mitochondrial cristae shape (Cogliati $et\ al$, 2013). To address whether loss of YME1L in the spinal cord affects cristae morphology, we analyzed axons in the dorso-lateral tract of the spinal cord by TEM. We did not detect abnormal mitochondria in asymptomatic NYKO mice at 6 weeks of age (Fig 6C and D). However, 17.9% of the mitochondria in the dorso-lateral tract were swollen and harbored disturbed cristae in 17-week-old NYKO mice (Fig 6C and D). The fraction of mitochondria with aberrant cristae increased further in aged, 31-week-old NYKO mice, which accumulated 22.2% abnormal mitochondria in the analyzed area (Fig 6C and D).

Thus, spinal cord mitochondria lacking YME1L show progressive deterioration of both respiratory activities and cristae structure. Although these deficiencies may contribute to axonal degeneration in aged mice, they become only apparent upon onset of axonal degeneration and therefore are unlikely to initiate the degenerative process.

---

**Figure 4.  Progressive axonal degeneration of dorso-lateral tracts and neuroinflammation in NYKO spinal cords.**

A  Spinal cord tracts anatomy, transverse section stained with toluidine blue. 1 = dorso-lateral tracts (ascending pathways), 2 = ventro-lateral tracts (ascending pathways), 3 = ventral tracts (descending pathways).

B  Transverse semithin sections of spinal cords of WT and NYKO mice of the indicated age. Sections were stained with toluidine blue. Orange arrows indicate degenerating neurons (myelin dense bodies, MDBs). Scale bars, 25 μm.

C  Myelin dense bodies (MDBs) in dorso-lateral tracts of 6- to 7-week-old WT ($n = 4$) and NYKO ($n = 4$), 17- to 18-week-old WT ($n = 4$) and NYKO ($n = 4$), and 31- to 32-week-old WT ($n = 6$–7) and NYKO ($n = 5$–6) mice spinal cords.

D  Immunoblots of 31- to 32-week-old spinal cord lysates. GAPDH was used to control for gel loading.

E  mRNA levels of proinflammatory cytokines from spinal cords of 31- to 32-week-old mice (WT, $n = 5$; NYKO, $n = 5$). Transcript levels were normalized to $Hprt$ mRNA levels.

Data information: Unpaired $t$-test, *$P \leq 0.05$, ***$P \leq 0.001$, ****$P \leq 0.0001$, ns = not significant. Data are means ± SEM.

Source data are available online for this figure.

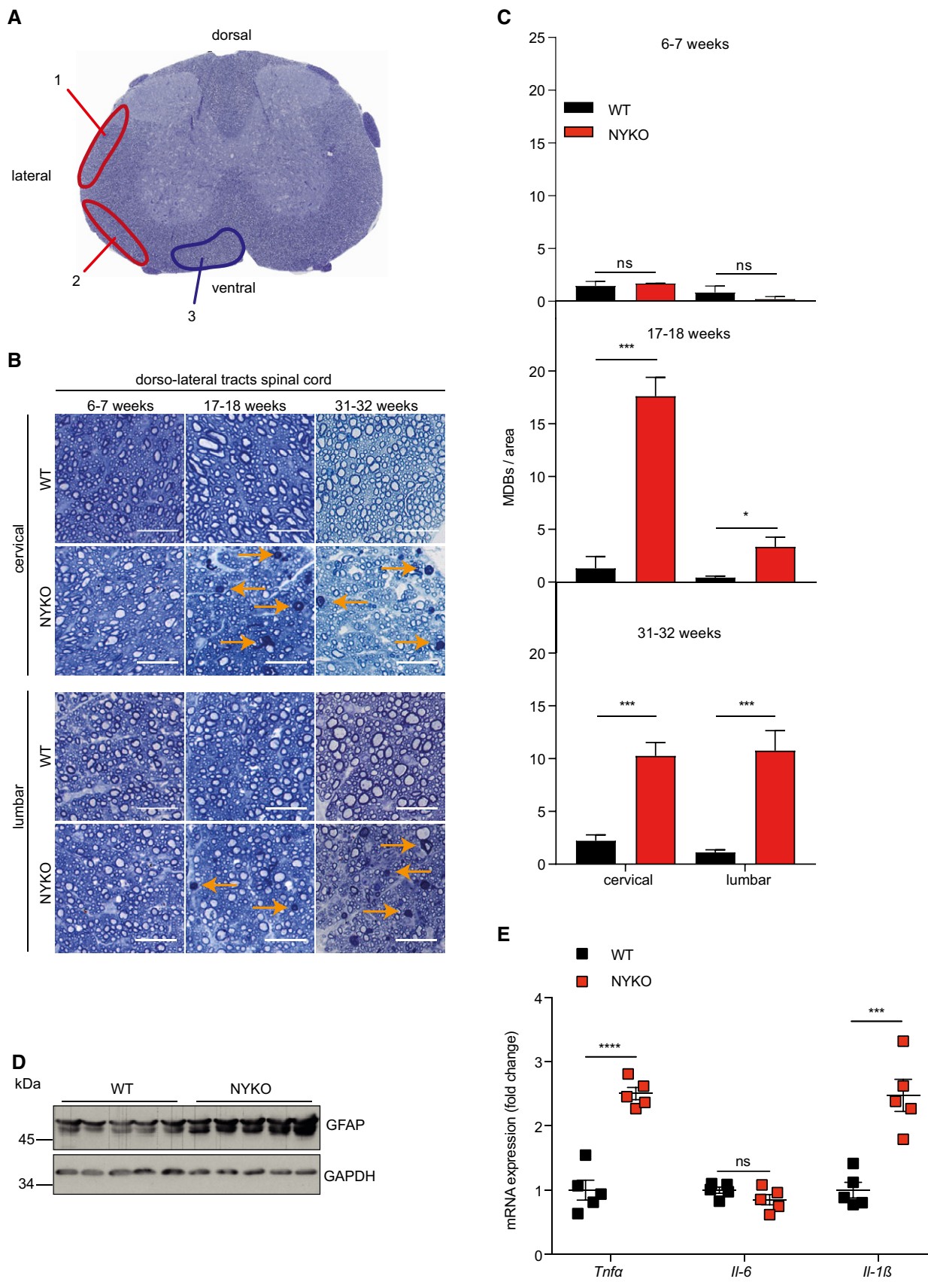

Figure 4.

## Mitochondrial fragmentation upon loss of YME1L in the nervous system

Disturbances of mitochondrial fusion or fission have been shown to correlate with mitochondrial trafficking defects (Sheng, 2014; Misgeld & Schwarz, 2017). Therefore, we examined the morphology of the mitochondrial network in spinal cord and brain of NYKO mice. We determined the mitochondrial length in individual axons using TEM of sagittal sections of the spinal cord of 6-week-old NYKO mice (Fig 7A and B). Mitochondria were significantly shorter (median 0.676 μm) than spinal cord mitochondria in control animals (median 0.879 μm) (Fig 7A and B). Similarly, we observed mitochondrial fragmentation in cerebella from 28-week-old NYKO and control mice by TEM analysis (Fig 7C and D). Moreover, visualization of cerebellar mitochondria by TOMM20 immunostaining revealed fragmentation of the mitochondrial network in the absence of YME1L (Appendix Fig S5A).

The loss of YME1L was found to impair the proteolytic cleavage of OPA1 and to thereby trigger fragmentation of the mitochondrial network in cultured cells and cardiomyocytes *in vivo* (Stiburek et al, 2012; Anand et al, 2014; Wai et al, 2015). We therefore monitored OPA1 processing in different brain regions and the spinal cord of 6- and 31-week-old NYKO and control mice (Fig 7E and Appendix Fig S5B). The S-OPA1 form d normally generated by YME1L was not detected in brains and spinal cords of NYKO mice, whereas the steady-state level of the S-OPA1 form c was increased (Fig 7E and Appendix Fig S5B). Consistently, L-OPA1 forms that are known to mediate mitochondrial fusion accumulated at decreased levels (Fig 7E and Appendix Fig S5B). Other components of the mitochondrial fusion and fission machineries accumulated at normal levels in spinal cords of 6-week-old NYKO mice (Appendix Fig S5C).

These experiments identify impaired OPA1 processing and mitochondrial fragmentation in the spinal cord as an early phenotype in NYKO mice that precedes disturbances in cristae morphogenesis and respiration. Notably, mitochondrial morphology was affected both in the brain and in the spinal cord but resulted only in the spinal cord in apparent axonal degeneration, suggesting that additional deficiencies may contribute to the degenerative phenotype. YME1L is a multifunctional peptidase with quality control function and its loss may therefore broadly affect mitochondrial proteostasis. Indeed, we observed the accumulation of other substrates of YME1L, such as the lipid transfer proteins PRELID1 and STARD7 or the protein translocase subunit TIMM17A in different brain regions and spinal cords of NYKO mice (Fig 7E and Appendix Fig S5B).

## OMA1 delays axonal degeneration in YME1L-deficient spinal cord

To directly assess the role of mitochondrial fragmentation for degenerative phenotypes in YME1L-deficient mice, we turned our attention to the second OPA1 processing peptidase OMA1. The loss of YME1L has been shown to activate OMA1 *in vitro* and *in vivo*, resulting in increased OPA1 processing, decreased accumulation of fusion-active, L-OPA1, and fragmentation of the mitochondrial network. Deletion of *Oma1* restored tubular mitochondria and suppressed cardiomyocyte death and heart failure in the absence of YME1L (Wai et al, 2015). We therefore crossed NYKO mice with *Oma1*^fl/fl mice (Baker et al, 2014) to obtain mice lacking YME1L (NYKO mice), OMA1 (NOKO mice), or both proteases (NYOKO mice) in the nervous system.

Immunoblot analysis of tissue lysates from retina and spinal cord from 6-week-old mice revealed stabilization of L-OPA1 in NYOKO mice when compared to NYKO mice while other YME1L substrates such as PRELID1, STARD7, and TIMM17A remained largely unaffected (Figs 8A and EV4A). Despite the loss of both OPA1 processing peptidases, we observed the formation of low levels of S-OPA1 forms in NYOKO mice, presumably reflecting the residual expression of YME1L and OMA1 in cells deriving from lineages other than astroglial or neuronal ones (Fig 8A). In agreement with the stabilization of L-OPA1, we observed tubular mitochondria with normal cristae morphology and a similar mitochondrial length in spinal cord axons of NYOKO mice when compared to control mice (Fig 8B and Appendix Fig S6A and B). Thus, deletion of *Oma1* suppressed mitochondrial fragmentation in the spinal cord by stabilizing L-OPA1.

Despite the accumulation of L-OPA1, YME1L-deficient mice developed microphthalmia and cataracts independent of the presence of OMA1 (Fig 8C). Retinal inflammation was not suppressed upon deletion of *Oma1*, as indicated by the accumulation of GFAP in retinal lysates of both NYKO and NYOKO mice but not of NOKO mice (Fig 8D). Moreover, the OPL of NYOKO mice appears disorganized and contained nuclei as observed in NYKO mice (Fig EV4B and C). The loss of OMA1 alone had no apparent effect on eye development (Fig 8C).

---

**Figure 5.  YME1L facilitates mitochondrial trafficking in neurons.**

A    Fluorescence images of isolated cortical neurons from *Yme1l*^fl/fl animals expressing CAG-GFP (WT) or CAG-Cre-IRES-GFP (*Yme1l*^−/−) at DIV12 and stained with TOMM20 to visualize mitochondria. Scale bars, 30 μm.

B–E    Quantification of total axon length (WT, n = 13 neurons; *Yme1l*^−/−, n = 13 neurons), total dendritic length (WT, n = 12 neurons; *Yme1l*^−/−, n = 13 neurons), axonal branch points (WT, n = 12 neurons; *Yme1l*^−/−, n = 13 neurons), and number of dendrites (WT, n = 12 neurons; *Yme1l*^−/−, n = 13 neurons). Individual neurons originate from two independent preparations.

F, G    Kymographs showing mitochondrial movements in representative axons. Vertical lines correspond to stationary mitochondria and diagonal lines to moving mitochondria. Mitochondria were stained using Mitotracker. In total, 39 axonal segments were analyzed for WT control neurons and 37 axonal segments for *Yme1l*^−/− neurons at DIV7.

H, I    Kymographs showing mitochondrial movements in representative axons. Vertical lines correspond to stationary mitochondria and diagonal lines to moving mitochondria. Mitochondria were stained using Mitotracker. In total, 120 axonal segments were analyzed for WT control neurons and 107 axonal segments for *Yme1l*^−/− neurons at DIV 10–14.

J    Axonal area occupied by mitochondria in *Yme1l*^−/− neurons relative to WT controls. In total, 120 axonal segments were analyzed for WT control neurons and 107 axonal segments for *Yme1l*^−/− neurons at DIV 10–14.

Data information: Unpaired *t*-test, ns = not significant, *$P \leq 0.05$. Data are means of three independent experiments ± SEM.
Source data are available online for this figure.

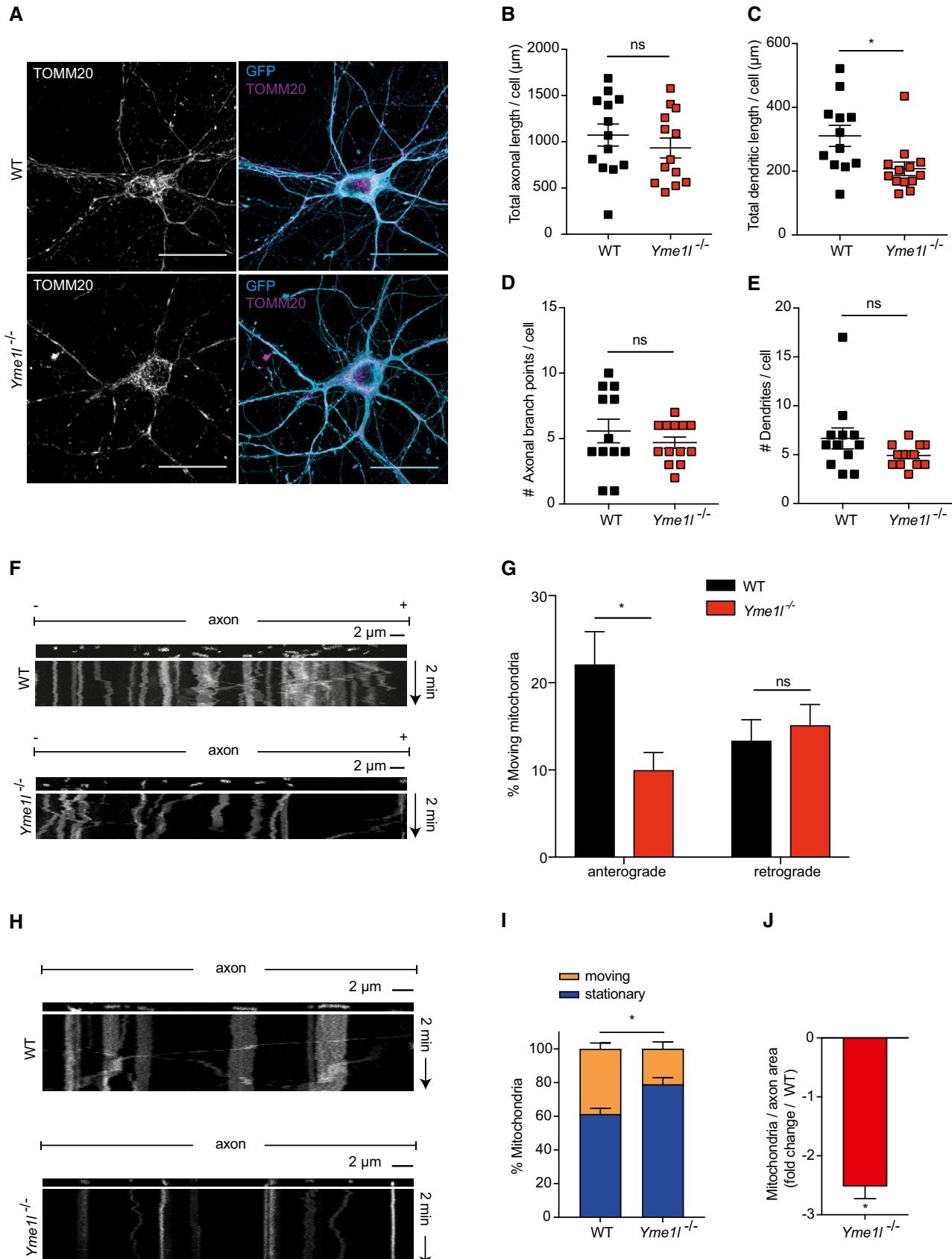

**Figure 5.**

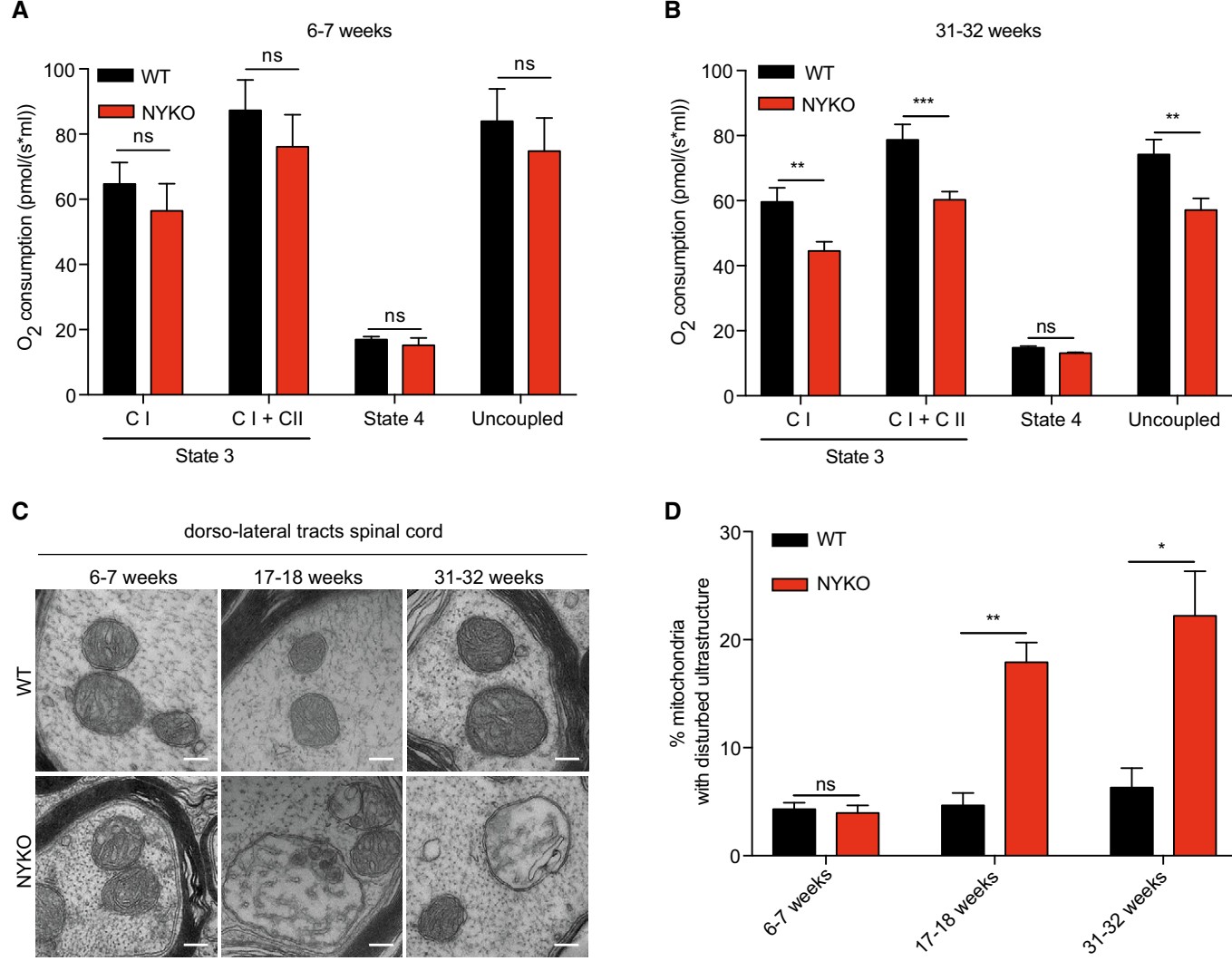

**Figure 6.  Late-onset respiratory and cristae defects in spinal cords of NYKO mice.**

A, B   Oxygen consumption rates of spinal cord mitochondria isolated from WT (*n* = 4) and NYKO (*n* = 4) mice in the presence of pyruvate–glutamate–malate (C I) and pyruvate–glutamate–malate + succinate (C I + C II) as substrates. State 3 (substrates + ADP), state 4 (+ oligomycin), uncoupled (+ FCCP).

C, D   TEM of dorso-lateral tracts of spinal cords of 6- to 7-week-old WT (*n* = 4, 1209 mitochondria) and NYKO (*n* = 4, 1792 mitochondria) mice, 17- to 18-week-old WT (*n* = 4, 442 mitochondria) and NYKO (*n* = 4, 724 mitochondria) mice, and 31- to 32-week-old WT (*n* = 3, 764 mitochondria) and NYKO (*n* = 3, 817 mitochondria) mice. Scale bars, 200 nm.

Data information: Unpaired *t*-test, *$P \leq 0.05$, **$P \leq 0.01$, ***$P \leq 0.001$, ns = not significant. Data are means ± SEM.

We analyzed brain and spinal cord of these mice in further experiments. Nissl stainings of different brain regions of 6-week-old NYOKO mice did not reveal apparent differences in NOKO and NYOKO mice when compared to control littermates (Appendix Fig S6C). We then assessed axonal degeneration in spinal cords of NYOKO mice monitoring the accumulation of MDBs (Fig 8E).

**Figure 7.  Defective proteostasis and accelerated OPA1 processing in brain and spinal cord of NYKO mice.**

A   TEM analysis of sagittal spinal cord sections of 6- to 7-week-old WT and NYKO mice. Scale bars, 500 nm.

B   Frequency distribution of mitochondrial lengths in spinal cords of 6- to 7-week-old WT (182 mitochondria) and NYKO mice (192 mitochondria). Kruskal–Wallis test, *$P < 0.05$.

C   TEM analysis of cerebellar sections of 28- to 32-week-old WT and NYKO mice. Scale bars, 1 μm.

D   Frequency distribution of mitochondrial aspect ratios were calculated from WT (*n* = 210) and NYKO (*n* = 230) mitochondria. Mann–Whitney test, ****$P < 0.0001$. Data are presented as median values.

E   Immunoblot analysis of forebrain, cerebellar and spinal cord lysates from 31- to 32-week-old WT and NYKO mice (*n* = 3) using the indicated antibodies. SDHA and LONP1 were used to control for gel loading. * indicates unspecific antibody binding.

Source data are available online for this figure.

◀▶

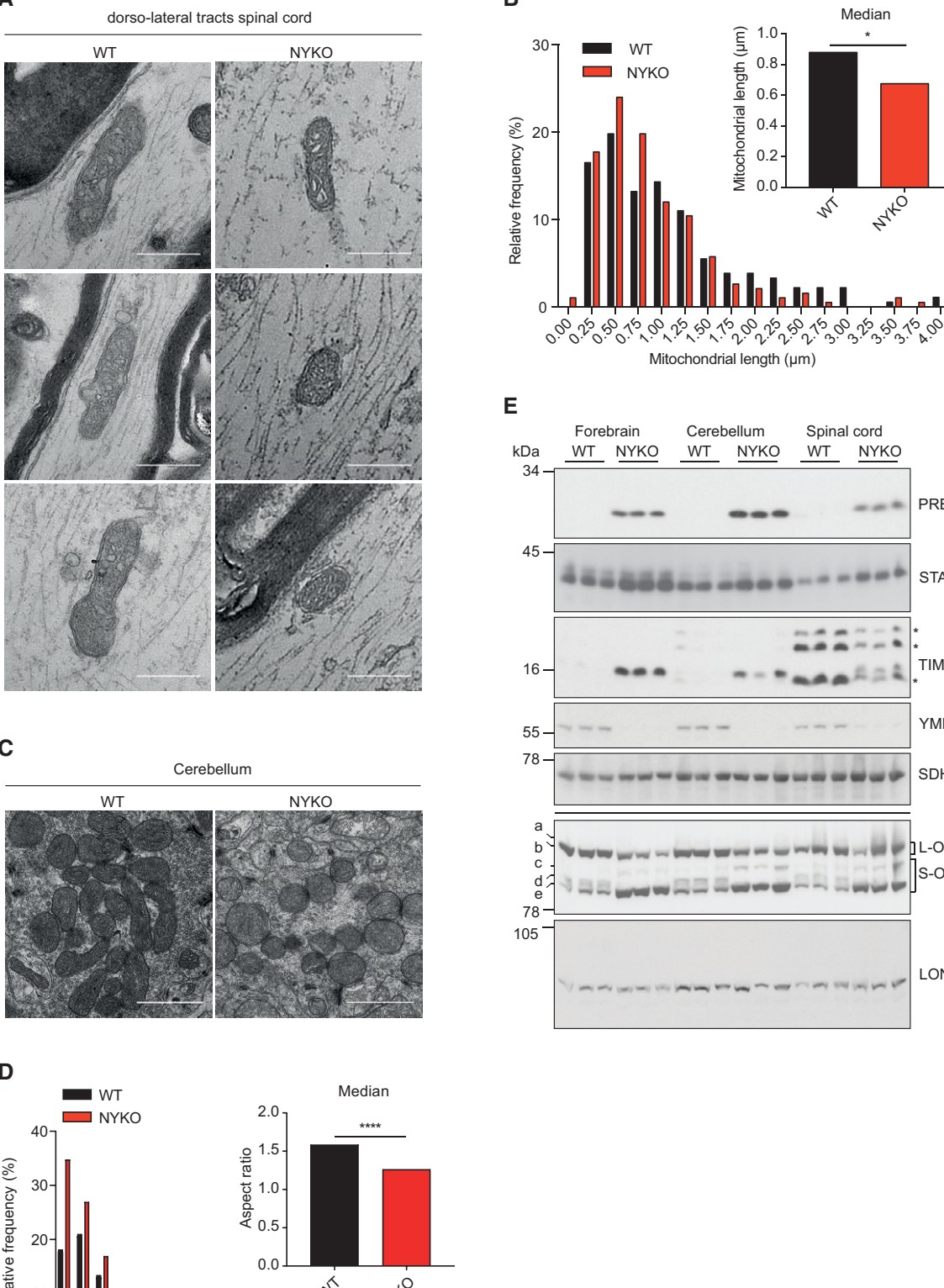

**Figure 7.**

Strikingly, while NYKO mice showed the first signs of axonal degeneration after 17–18 weeks (Fig 4B and C), we observed axonal degeneration of spinal cord neurons in NYOKO mice already at 6–7 weeks (Fig 8E and F). MDBs accumulated massively both in cervical and in lumbar regions of the dorso-lateral tracts (Fig 8F). The accumulation of MDBs in the spinal cord of these mice was not restricted to the dorso-lateral tracts but occurred also in ventro-lateral and ventral tracts (Fig EV4D and E), as it has been also observed in aged NYKO mice (Fig EV2C and D). On the other hand, we did not detect MDBs at significantly increased levels in the spinal cord of NOKO mice lacking only OMA1 (Fig 8E and F). Furthermore, we observed increased expression of proinflammatory cytokines in the spinal cord of NYOKO mice when compared to control mice (Fig EV4F). Neuroinflammation and the accelerated degeneration of spinal cord axons in NYOKO mice correlated with deficiencies in mitochondrial transport observed in primary cortical neurons *in vitro* lacking both proteases, YME1L and OMA1 (Fig EV5A and B). In agreement with these observations, walking beam tests revealed a severe locomotor impairment of NYOKO mice already at 6 weeks of age (Fig 8G), that is, at an age when NOKO and NYKO mice are asymptomatic (Fig 2D and 8G).

To address whether stalled macroautophagy could be causative for the worsening of the phenotype in NYOKO mice, we analyzed spinal cords isolated from 6- to 7-week-old mice by immunoblotting and quantitative real-time PCR. The autophagy marker proteins p62 and LC3B did not accumulate at different levels when comparing spinal cord protein lysates from mice depleted for YME1L, OMA1, or both proteases (Appendix Fig S6D). Moreover, we did not observe differences in the mRNA expression of autophagy-related genes when comparing NYOKO mice spinal cords to control tissue (Appendix Fig S6E). In addition, immunoblot analysis of different mitochondrial marker proteins revealed no effect on overall mitochondrial mass between the different genotypes (Appendix Fig S6F).

Together, these experiments demonstrate that the loss of OMA1 does not suppress ocular deficiencies present in NYKO mice and deteriorates axonal degeneration in the spinal cord of these mice, although mitochondrial morphology was restored.

## Discussion

The *i*-AAA protease YME1L serves as a quality control enzyme in mitochondria and regulates membrane dynamics and intramitochondrial lipid trafficking. Despite these housekeeping functions and its ubiquitous expression, we demonstrate that the loss of YME1L in the nervous system of the mouse causes striking cell-type-specific deficiencies: Young mice show microphthalmia, cataracts, and retinal inflammation followed by the development of progressive axonal degeneration specifically in the dorso-lateral column of the spinal cord. This results in impaired locomotor activity and hind limb paralysis of the mice starting at approximately 17 weeks of age. Our loss-of-function model for YME1L in the nervous system therefore recapitulates some clinical features of a neuromuscular disorder that is caused by homozygous missense mutations in *YME1L* and characterized by developmental delay, ocular dysfunction, ataxia and athetotic and stereotypic movements

(Hartmann *et al*, 2016). Phenotypic differences between NYKO mice and human patients are likely explained by the hypomorphic character of the missense mutation in human (Hartmann *et al*, 2016). Our results also uncover an unexpected role of YME1L for the development of the eye and point to a novel pathway triggering microphthalmia and cataracts in mitochondrial diseases, which is independent of respiratory dysfunction and highlights the importance of mitochondrial proteostasis for axonal maintenance in neurodegenerative disorders.

The dorso-lateral tract of the spinal cord harbors ascending neurons that relay proprioceptive signals from the hind limbs to the cerebellum (Hantman & Jessell, 2010). The loss of these neurons thus likely explains movement deficiencies in NYKO mice, which show phenotypic similarities to mice lacking the *m*-AAA protease subunit paraplegin (SPG7) mutated in hereditary spastic paraplegia (HSP7) (Ferreirinha *et al*, 2004). It should be noted, however, that we cannot exclude that degeneration of distal portions of motor axons or synapses contributes to the degenerative phenotype. Disturbed proprioception produces ataxia, as exemplified by Friedreich ataxia, an autosomal recessive disorder caused by mutations in mitochondrial frataxin (Burk, 2017). Loss of frataxin in disease leads to atrophy of root ganglion cells and affects dorsal and ventral tracts of the spinal cord. Moreover, axonal degeneration in the spinal cord is commonly observed in spinocerebellar ataxias in combination with cerebellar atrophy (Fratkin & Vig, 2012). The pathological phenotype of NYKO mice is thus reminiscent of other mitochondrial disorders, although different areas of the spinal cord may be affected. Notably, we observed a reduced gain of body weight similar to other mouse models for mitochondriopathies (Kruse *et al*, 2008; Wang *et al*, 2016). It is possible that deletion of *Yme1l* in the nervous system affects the feeding behavior via a direct effect on hunger and satiety controlling neurons. Alternatively, the reduced body weight may be indirectly caused by the neuronal pathology of the mice.

Interestingly, also congenital microphthalmia has been described as a pathological condition in a rare mitochondrial disease involving OXPHOS dysfunction and activation of a non-canonical cell death pathway (Indrieri *et al*, 2012, 2013). It should be noted, however, that disturbed proteostasis in NYKO mice likely causes microphthalmia and cataracts via an independent pathway than described for these models. The degenerative phenotype of NYKO mice is distinct from mouse models of OXPHOS dysfunction, which show optic neuropathy characterized by the loss of retinal ganglion cells combined with brain atrophy (Fukui *et al*, 2007; Kruse *et al*, 2008; Quintana *et al*, 2010; Yu-Wai-Man *et al*, 2016). In contrast, respiratory defects become apparent only in aged NYKO mice. This model manifests an impaired eye development associated with microphthalmia already at young age, with axons of retinal ganglion cells being largely unaffected. Moreover, NYKO mice show a late-onset progressive axonal degeneration in the dorso-lateral tracts of the spinal cord, whereas a general brain atrophy does not manifest. NYKO mice thus represent a novel model for microphthalmia and cataracts independent of OXPHOS deficiencies.

We did not detect myelin loss at early stages, suggesting that cell autonomous mechanisms in neurons rather than demyelination trigger axonal degeneration in NYKO mice. Synaptic activity and neuronal survival depend on the transport of mitochondria

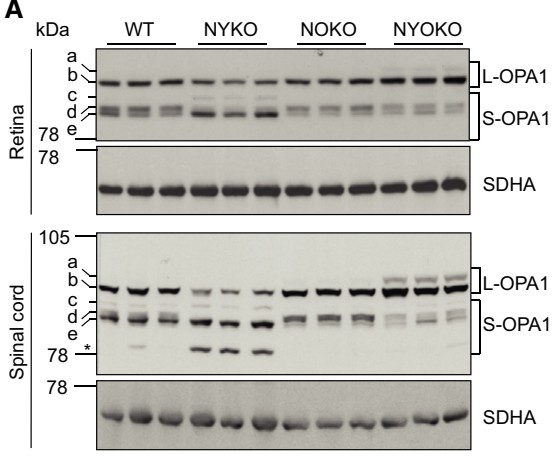

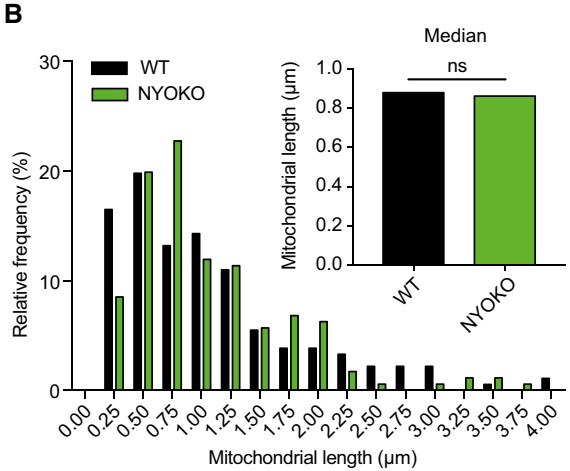

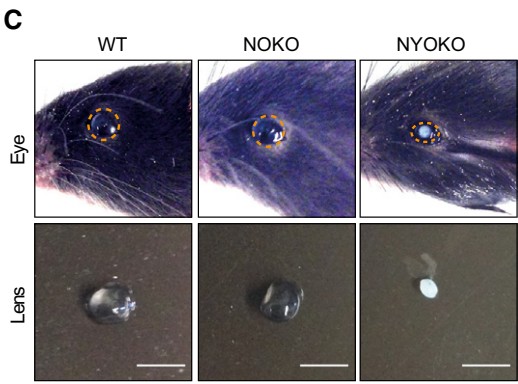

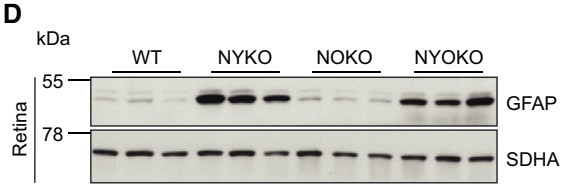

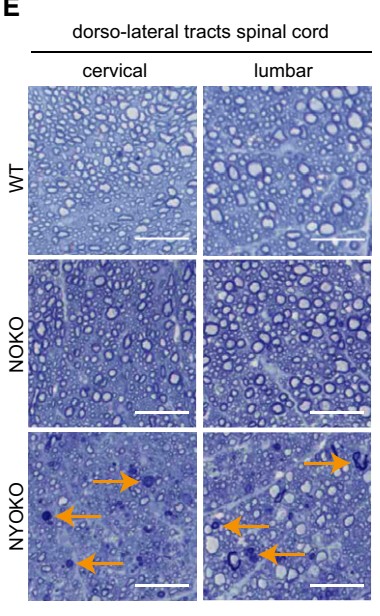

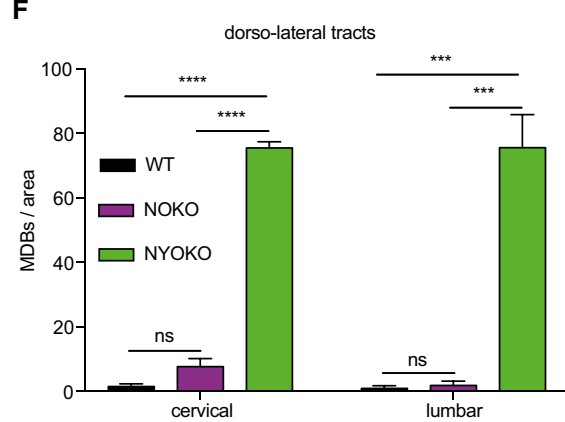

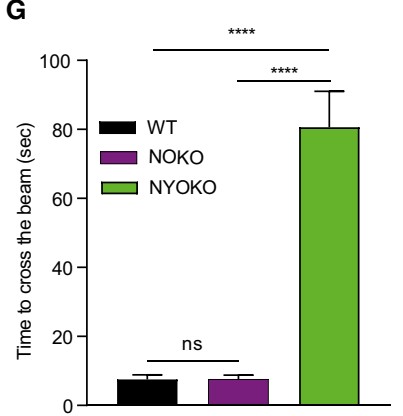

**Figure 8.**

◀

**Figure 8.  Loss of OMA1 does not restore ocular dysfunction and deteriorates axonal degeneration in the spinal cord of NYKO mice.**

A   Immunoblot analysis of retinal and spinal cord lysates from 6- to 7-week-old WT, NYKO, NOKO, and NYOKO mice (*n* = 3) using the indicated antibodies. SDHA was used as a loading control. * was not reproducibly detectable in all experiments.

B   Frequency distribution of mitochondrial lengths in spinal cords of 6- to 7-week-old NYOKO mice (176 mitochondria) compared to WT (as in Fig 7B). Kruskal–Wallis test, ns = not significant.

C   Representative images of eyes and lenses from 6- to 7-week-old WT and mice lacking OMA1 (NOKO) or both YME1L and OMA1 (NYOKO) in the nervous system. Orange dashed lines mark eye morphology. Scale bars, 2.5 mm.

D   Immunoblot analysis of retina lysates of 6- to 7-week-old WT, NYKO, NOKO, and NYOKO mice (*n* = 3). SDHA was used to control for gel loading.

E   Transverse semithin sections of spinal cords of 6- to 7-week-old WT, NOKO, and NYOKO mice. Sections were stained with toluidine blue. Orange arrows point to MDBs indicating degenerating neurons. Scale bars, 25 μm.

F   MDBs in dorso-lateral tracts of 6- to 7-week-old WT (*n* = 3), NOKO (*n* = 3), and NYOKO (*n* = 3) mice.

G   Walking beam test of 6- to 7-week-old WT (*n* = 6), NOKO (*n* = 5), and NYOKO (*n* = 7) mice.

Data information: Data were analyzed using one-way ANOVA. ***$P \leq 0.001$, ****$P \leq 0.0001$, ns = not significant. Data are means ± SEM.

Source data are available online for this figure.

and local energy supply (Lin & Sheng, 2015; Misgeld & Schwarz, 2017). Indeed, we observed deficiencies in mitochondrial trafficking in cultured neurons lacking YME1L, indicating that depletion of mitochondria from neurites may contribute to neuronal loss. In agreement with this hypothesis, mitochondrial transport deficiencies upon loss of the mitochondrial Rho GTPase 1 (Miro1) affect dendritic complexity and lead to secondary neuronal degeneration in the spinal cord before brain regions become affected (Nguyen *et al*, 2014; Lopez-Domenech *et al*, 2016). It should be noted that deletion of *Yme1l* does not profoundly affect mitochondrial respiration and only leads to respiratory defects at late stages. Similarly, the loss of YME1L impaired cristae morphogenesis only in aged mice. A general energy crisis in neurons is therefore unlikely to cause mitochondrial trafficking defects and axonal degeneration.

One of the earliest defects observed in YME1L-deficient neurons of the spinal cord is the fragmentation of the mitochondrial network, which becomes apparent already in 6-week-old NYKO mice. Disturbances in mitochondrial morphology, caused by either impaired fusion or fission, can cause deficiencies in the axonal trafficking of mitochondria (Sheng, 2014). Inhibition of mitochondrial fusion and mitochondrial fragmentation upon loss of MFN2 or OPA1 impairs axonal transport of mitochondria in cultured neurons and leads to neurodegeneration (Baloh *et al*, 2007; Bertholet *et al*, 2013). Similarly, depletion of axonal mitochondria in midbrain dopamine neurons was observed in the absence of DRP1 and mitochondrial fission (Berthet *et al*, 2014; Sheng, 2014; Gao *et al*, 2017). YME1L regulates mitochondrial dynamics via proteolytic cleavage of OPA1. Consistently, we have observed imbalanced OPA1 processing in YME1L-deficient spinal cord neurons: While OPA1 cleavage generating S-OPA1 form d is inhibited, L-OPA1 mediating mitochondrial fusion is present at decreased levels due to accelerated processing by OMA1.

Imbalanced OPA1 processing and mitochondrial fragmentation upon cardiac deletion of *Yme1l* are associated with cardiomyocyte death and heart failure, which can be suppressed by additional deletion of *Oma1* (Wai *et al*, 2015). *Oma1* deficiency also protects against mitochondrial fragmentation and acute ischemic kidney injury (Xiao *et al*, 2014). Similarly, loss of the membrane scaffold PHB2 induces OPA1 processing and mitochondrial fragmentation in neurons resulting in neurodegeneration in the brain, which can be delayed by deletion of *Oma1* (Korwitz *et al*, 2016). In agreement with these observations, ablation of *Oma1* stabilized L-OPA1 in the

spinal cord of NYKO mice and promoted elongation of mitochondria, consistent with increased mitochondrial fusion. Intriguingly, it did not restore axonal transport of mitochondria *in vitro* nor promote neuronal survival *in vivo*. Rather, neurodegeneration became already apparent in dorso-lateral tracts of 6-week-old mice and was also observed in ventro-lateral and ventral tracts of the spinal cord. We therefore conclude that accelerated OPA1 processing and mitochondrial fragmentation alone do not cause axonal degeneration in the spinal cord of NYKO mice. The absence of general brain atrophy despite the mitochondrial network being fragmented suggests that mitochondrial fragmentation *per se* is insufficient to cause broad neurodegeneration and that in related disorders additional defects should be considered (Burte *et al*, 2015). Moreover, these findings point to a novel, pro-survival function of OMA1 in the absence of YME1L. The accumulation of other OMA1 substrates may explain the worsening of the degenerative phenotype in NYOKO mice. Moreover, in view of the central role of mitochondrial proteostasis for neuronal survival (Sorrentino *et al*, 2017), it is conceivable that the concomitant loss of two proteases in the IMS with at least partially overlapping functions causes a collapse of mitochondrial proteostasis, which leads to premature axon degeneration (Misgeld & Schwarz, 2017).

Our experiments uncover strikingly different consequences of the loss of YME1L and imbalanced mitochondrial dynamics in different tissues: While deleterious in cardiomyocytes, mitochondrial fragmentation upon loss of YME1L alone does not cause general neuronal death, neither in the retina nor in the spinal cord or in various regions of the brain. How can this difference in cellular vulnerability be explained? The dynamic behavior of mitochondria is intimately linked to the cellular metabolism, and, in adaptation to various physiological demands, the morphology of the mitochondrial network can vary tremendously between different tissues (Mishra & Chan, 2016; Wai & Langer, 2016). Thus, different metabolic profiles including different fuel utilization may determine how disturbances in balanced mitochondrial dynamics affect cell survival in the absence of YME1L. Moreover, the function of YME1L in shaping the mitochondrial proteome may vary between tissues and cells depending on the expression of specific substrates. Regardless, the unexpected roles of YME1L specifically in the retina and in maintaining axonal integrity of spinal cord neurons involved in proprioception may provide the unique opportunity to study neuron-specific consequences of a disturbed mitochondrial proteostasis for eye development and movement coordination.

# Materials and Methods

### Mouse breeding and behavioral tests

All animal procedures were carried out in accordance with European, national and, institutional guidelines and were approved by local authorities (Landesamt für Natur, Umwelt, und Verbraucherschutz Nordrhein-Westfalen, Germany; approval number 84-02.04.2015.A313 and 84-02.05.40.14.066, 4.17.019). To obtain nervous system-specific NYKO, NOKO, and NYOKO mice, $Yme1l^{fl/fl}$, $Oma1^{fl/fl,}$ or $Yme1l/Oma1^{fl/fl}$ mice (Anand $et$ $al$, 2014; Wai $et$ $al$, 2015) were crossed to Nestin-Cre mice (Tronche $et$ $al$, 1999). $Yme1l^{fl/fl}$ or $Yme1l/Oma1^{fl/fl}$ Cre-negative littermates were used as wild-type (WT) controls. To exclude phenotypic consequences of Cre recombinase expression, heterozygous NYKO mice (het NYKO, $Yme1l^{fl/wt}$ Cre-positive) were analyzed. Unless otherwise indicated, groups included male and female C57BL/6 animals.

For the walking beam test, mice were trained to walk on a 90-cm-long and 3-cm-wide beam, elevated by 30 cm on a metal support, for three times on two consecutive days. The test was performed by allowing the mice to walk on a 1-cm-wide beam on the third day. The time they needed to cross the beam and the number of foot slips were measured per run. The wire mesh grip strength test was performed by placing the animals on a metal wire mesh above the cage. Following inversion of the wire mesh, the latency to fall was recorded three times per animal with a 5-min break in between the trials.

### Metabolic mouse monitoring

Body composition was determined $in$ $vivo$ with a nuclear magnetic resonance analyzer (minispec mq7.5, Bruker). For metabolic measurements, mice were acclimated for 1–2 days and then monitored for 48 h using a PhenoMaster System (TSE Systems) fitted with indirect calorimetry and activity monitors to measure activity and energy expenditure. Due to locomotor dysfunction of NYKO mice, food was provided from the bottom of the cage during the measurements for both genotypes.

### Tissue collection

Mice were deeply anesthetized with xylazine/ketamine and perfused transcardially with PBS and 4% paraformaldehyde (PFA). Tissues were dissected and post-fixed in 4% PFA for histology and immunofluorescence. For transmission electron microscopy (TEM), samples were post-fixed in 2% glutaraldehyde in 0.12 M phosphate buffer. Samples for protein and RNA extraction were taken after cervical dislocation and immediately frozen in liquid nitrogen.

### Retina histology

Retinal tissues were post-fixed in 4% PFA, embedded in paraffin, and cut into 5-μm sections. Afterwards sections were deparaffinized in xylol and rehydrated. After consequent hematoxylin and eosin staining, sections were dehydrated and scanned using a slide scanner (SCN400, Leica). For assessment of retinal disorganization, nuclei in the outer plexiform layer were quantified per 1,000 μm² in a blinded fashion on three different sections per animal.

### Brain histology

Brain tissues were post-fixed in 4% PFA and embedded in 6% agar, and 40-μm sections were cut using a vibratome (VT1200S, Leica). For Nissl staining, sections were incubated in thionine solution before being dehydrated and imaged using a slide scanner (SCN400, Leica). For immunofluorescence, free-floating sections were permeabilized and blocked in 0.4% (v/v) Triton X-100 and 10% (v/v) goat serum in PBS for 1 h at room temperature (RT). Primary antibodies were incubated overnight at 4°C. On the next day, fluorescent secondary antibodies were incubated for 2 h at RT. Finally, sections were mounted in FluorSave Reagent (Calbiochem). Primary antibodies were anti-TOMM20 (sc-11415, Santa Cruz Biotechnology Inc., 1:1,000), anti-IBA1 (019-19741, Wako, 1:500), anti-Calbindin (Calbindin D-28k 300, Swant, 1:500). Secondary antibodies were purchased from Invitrogen and diluted 1:1,000. Fluorescent images were acquired using a confocal microscope equipped with a 10× (NA 0.3) or 63× (NA 1.4, oil) objective lens (LSM 510 META, Zeiss). For assessment of mitochondrial network morphology, 1,600 μm² from 63× cerebellar images was quantified per animal in a blinded fashion.

### Semithin section and electron microscopy

The optic nerve and spinal cord tissues were post-fixed in 2% glutaraldehyde (Electron Microscopy Sciences) in 0.12 M phosphate buffer and treated with 2% osmium tetroxide (Electron Microscopy Sciences). After dehydration using ethanol and propylene oxide, tissues were embedded in Epon (Sigma). 500-nm semithin sections were cut using an ultramicrotome (EM UC7, Leica) and stained with 1% toluidine blue. Sections were scanned using an automated slide scanner (SCN400, Leica). For electron microscopy, 70-nm ultrathin sections were cut from Epon blocks and stained with uranyl acetate (Plano GMBH) and lead nitrate (Sigma). Images were acquired using a transmission electron microscope (JEM-2100 Plus, Jeol) equipped with Gatan ONE View camera. The g-ratio was determined by measuring the ratio between the diameter of the axon and the diameter of the myelinated fiber on individual electron micrographs from three mice per genotype. Mitochondrial aspect ratios, length, and the percentage of mitochondria with disturbed cristae were measured on individual electron micrographs from at least three mice per genotype. To quantify the formation of MDBs, the total amount of MDBs was determined per image and normalized to an area of 40,000 μm² from at least three mice per genotype. All quantifications were done in a blinded fashion using ImageJ (National Institutes of Health, Bethesda, United States).

### Immunoblotting

Tissues were homogenized in ice-cold RIPA buffer (50 mM Tris–HCl, pH 7.4, 150 mM NaCl, 1% Triton X-100, 0.1% SDS, 0.05% sodium deoxycholate, 1 mM EDTA, and complete protease inhibitor cocktail mix) using Precellys 24 (Bertin instruments). SDS–PAGE was used to separate 50–100 μg total protein, followed by transfer to nitrocellulose membranes, and immunoblotting using the following antibodies: rabbit anti-GFAP (Z0334, Dako, 1:1,000), mouse anti-GAPDH (sc-32233, Santa Cruz Biotechnology Inc., 1:1,000), mouse anti-PRELID1 (H00027166-M01, Abnova, 1:1,000), rabbit anti-STARD7

(15689-1-AP, Protein Tech Group, 1:2,000), rabbit anti-YME1L (11510-1-AP, Protein Tech Group, 1:1,000), rabbit anti-TIMM17A (GTX108280, GeneTex, 1:1,000), mouse anti-OPA1 (612606, BD Biosciences, 1:500), mouse anti-SDHA (459200, Invitrogen, 1:10,000), rabbit anti-LONP1 (HPA002192, Sigma, 1:1,000), mouse anti-cytochrome c (556433, BD Biosciences, 1:1,000), rabbit anti-caspase-8 (4790, Cell Signaling Technology, 1:1,000), mouse anti-caspase-9 (9508, Cell Signaling Technology, 1:1,000), mouse anti-NDUFA9 (459100, Invitrogen, 1:1,000), mouse anti-UQCRC2 (459220, Invitrogen, 1:1,000), mouse anti-ATP5A (459240, Invitrogen, 1:1,000), mouse anti-MTCO1 (459600, Invitrogen, 1:1,000), mouse anti-DRP1 (611112, BD Biosciences, 1:1,000), rabbit anti-pDRP1 (S616, 3455, Cell Signaling, 1:1,000), rabbit anti-MFF (17090-1-AP, Protein Tech Group, 1:1,000), mouse anti-HSP60 (ADI-SPA-806-D, Enzo, 1:5,000), rabbit anti-MFN2 (ab50838, Abcam, 1:1,000), rabbit anti-MID51 (20164-1-AP, Protein Tech Group, 1:1,000), rabbit anti-FIS1 (10956-1-AP, Protein Tech Group, 1:1,000), mouse anti-actin (A5441, Sigma, 1:40,000), mouse anti-p62 (H00058472-B01P, Abnova, 1:500), rabbit anti-LC3B (L7543, Sigma, 1:1,000), rabbit anti-TOMM20 (sc-11415, Santa Cruz Biotechnology Inc., 1:1,000). Horizontal lines in Figures separate individual gels using the same samples.

### qRT–PCR

Total RNA was extracted using TRIzol Reagent (Life Technologies) and NucleoSpin RNA columns (Macherey-Nagel). cDNA was synthesized with the SuperScript III First-Strand Synthesis System kit (Invitrogen). qRT–PCR was performed with Power SYBR Green PCR Master Mix (Applied Biosystems) with *Hprt* as a reference gene. Data were analyzed according to the ΔCT method. For details on primer sequences, see Appendix Primer Sequences.

### Isolation of cortical neurons and mitochondrial trafficking experiments

Cortices were isolated from *Yme1l*^fl/fl^ or *Yme1l*^fl/fl^/*Oma1*^fl/fl^ embryos between embryonic day 13 and 16.5 in ice-cold HBSS (Invitrogen) supplemented with 10 mM HEPES-KOH. To isolate neurons, cortices were isolated in DMEM (Invitrogen) with glutamine in the presence of papain (20 U, Sigma) for 20 min followed by mechanical dissociation. Papain activity was stopped by adding fresh DMEM supplemented with 10% FBS. Dissociated neurons were electroporated with either CAG-Cre-IRES-GFP or CAG-GFP-expressing plasmids (Motori *et al*, 2013) using an Amaxa Nucleofector and nucleofection reagents (Lonza). Afterwards neurons were plated on poly-D-lysine (Sigma) pre-coated glass coverslips at a density of $10^5$ cells per coverglass and maintained in Neurobasal added with B27 and Glutamax (Invitrogen). By DIV7 or 10–14, cells were incubated with Mitotracker red (75 nM, ThermoFisher) for 30 min in imaging solution (124 mM NaCl, 10 mM D-glucose, 10 mM HEPES-KOH (pH 7.3), 3 mM KCl, 2 mM CaCl$_2$, 1 mM MgCl$_2$, pH 7.4), followed by two washing steps. Live imaging was performed on a Leica SP8 laser-scanning confocal microscope equipped with a white light laser, hybrid detectors, 63× (1.4 NA) oil objective, and heating stage. Glass coverslips were transferred to a recording chamber containing imaging media and equilibrated for 5 min at 37°C prior to imaging. Images were acquired at a frame rate of 0.8 Hz (1 frame

every 1.3 s) for 2 min. Simultaneous laser excitation of GFP and Mitotracker Red was achieved using wavelengths of 488 and 546 nm, respectively, and by adjusting the interval of detection to avoid signal cross-talk. Laser excitation was maintained in the range of 1–3% of maximum power to prevent phototoxicity.

Mitochondrial motility was analyzed using ImageJ (National Institutes of Health, Bethesda, United States) by normalizing the number of mitochondria moving to the total number of mitochondria present in axons of the acquired images. Identification of axonal (versus dendritic) mitochondria in live-imaging experiments was achieved by first assessing neuronal morphology in GFP and Cre-GFP-expressing neurons. GFP signal (labeling the whole morphology of analyzed neurons) was first acquired to unambiguously identify axonal fibers. According to the age of neurons, axonal identity was defined as (i) processes emerging from the body and extending in length three times or longer than any other primary process, (ii) presence of *en-passant* boutons and/or growth cones at their terminals, and (iii) absence of dendritic protrusions. To facilitate the analysis of axonal mitochondria in older cultures (DIV 10–14), fields of acquisitions were chosen to be distant from any cell bodies and dendrites, in areas specifically enriched by axonal fibers visibly decorated by boutons. Mitochondria were classified as moving if they moved at least 2 μm during the entire duration of the acquisition otherwise they were categorized as stationary. Kymographs were created using Multiple Kymograph plugin in ImageJ. The density of mitochondria in fibers was calculated by superimposing Mitotracker images on an image mask of the GFP-expressing fibers. Only GFP-positive fibers were considered for analysis.

### Immunocytochemistry cortical neurons

Cortical neurons were isolated and cultured as described above. Upon fixation with 4% paraformaldehyde, cells were permeabilized using 1% Triton X-100 and blocked in 3% BSA in PBS. Antibody incubation was performed overnight at 4°C. The following antibodies were used for immunofluorescence: chicken anti-GFP (GFP-1020, Aves Labs Inc., 1:1,000), mouse anti-SMI-312 (ab24574, Abcam, 1:1,000), rabbit anti-TOMM20 (sc-11415, Santa Cruz Biotechnology Inc., 1:1,000), guinea pig anti-vGat (131004, Synaptic Systems, 1:1,000), mouse anti-vGlut1 (135311, Synaptic Systems, 1:1,000). Secondary antibodies were purchased from Jackson ImmunoResearch.

### Oxygraph measurements

To determine oxygen consumption rates with Oxygraph-2k (Oroboros Instruments), mitochondria were isolated from mouse spinal cords. To this end, dissected spinal cord tissues were homogenized in a standard homogenization buffer (20 mM HEPES-KOH, pH 7.4, 220 mM D-mannitol, 70 mM sucrose, 2 mM EGTA, 0.1% BSA (w/v), 1× complete protease inhibitor (Roche)) using a Dounce Teflon homogenizer (PotterS, Braun) at 100 *g* with 10 strokes followed by differential centrifugation. To measure mitochondrial complex I-dependent respiration, ADP (2 mM), pyruvate (5 mM), malate (2 mM), and glutamate (20 mM) were added to mitochondria (150 μg) in respiration buffer (2 ml, 120 mM sucrose, 50 mM KCl, 20 mM Tris–HCl, 1 mM EGTA, 4 mM KH$_2$PO$_4$, 2 mM MgCl$_2$, 0,1% (w/v) BSA). Combined complex I and complex II respiration was

assessed by addition of succinate (10 mM). Subsequently, mitochondrial coupling was evaluated by the inhibition of ATP synthase by adding 1.5 μg/ml oligomycin (1.5 μg/ml) and uncoupling by a multiple-step carbonylcyanide p-trifluoromethoxyphenyl-hydrazone (FCCP) titration.

**Blue native gel electrophoresis (BN-PAGE)**

Spinal cord mitochondria (100–200 μg) were solubilized in 50 mM NaCl, 5 mM 6-aminohexanoic acid, 50 mM imidazole–HCl, pH 7, 10% (v/v) glycerol, and 50 mM $KP_i$-buffer, pH 7.4, supplemented with detergent (6 g digitonin/g protein or 2.5 g DDM/g protein). After extraction for 30 min and a clarifying spin at 16,000 $g$ for 20 min, samples were separated by 3–13% BN-PAGE.

**Statistical analyses**

Statistical analyses were performed using GraphPad PRISM (version 7.0a). Data sets with only two independent groups were analyzed for statistical significance using unpaired two-tailed Student's $t$-test. Data sets with more than two groups were analyzed using one-way analysis of variance (ANOVA). All displayed values are presented as mean ± SEM, if not stated otherwise. All $P$-values below 0.05 were considered significant with *$P \leq 0.05$, **$P \leq 0.01$, ***$P \leq 0.001$, ****$P \leq 0.0001$. For details on statistics and individual $P$-values, see Appendix Statistical Analyses. We have not performed any specific tests to estimate variation or outliers within each group of data. No statistical method was used to predetermine sample size. The sample size included at least three biological replicates, if not stated otherwise. We have not excluded any samples from our studies. The reasons for exclusion would be: wrong genotype of animals; sample was obviously degraded or in some way compromised. We have used an unbiased approach in choosing animals for our experiments. Animals from different litters coming from different parents were used in all experiments. If possible, data analysis was performed blinded.

Expanded View for this article is available online.

## Acknowledgements
We thank Steffen Hermans for the PhenoMaster analysis, Anne Korwitz for help in immunohistochemistry, and the CECAD Imaging Facility for excellent support. We are grateful to the department of Thomas Langmann, University Clinic Cologne, for helpful discussion. This work was supported by grants of the Deutsche Forschungsgemeinschaft SFB1218 A1; La918/and the Max-Planck-Society to T.L and EMBO fellowships to M.P and T.M (ALTF 649-2015, LTFCO-FUND2013, GA-2013-609409, ALTF 1220-2014). T.M was also supported by an Alexander von Humboldt Postdoctoral Fellowship. M.B was supported by grants of the Deutsche Forschungsgemeinschaft SFB1218 A7 and ERC StG 2015 (grant number 677844).

## Author contributions
H-GS, TW, MB, EIR, and TL contributed to the study design. H-GS, GW, AH, TK, MP, TM, SA, and EB performed experiments. H-GS and TL wrote the manuscript. All authors discussed and commented on the manuscript.

## Conflict of interest
The authors declare that they have no conflict of interest.

**The paper explained**

**Problem**

Disturbances in mitochondrial functions are associated with various neurological disorders, which are often highly cell-type-specific. Mutations in the $i$-AAA protease YME1L that regulate mitochondrial proteostasis and dynamics cause a mitochondriopathy in human with ocular dysfunction and movement deficiencies, but pathogenic mechanisms are not understood.

**Results**

We developed a nervous system-specific mouse model for YME1L deficiency. These mice show striking cell-specific deficiencies: They present with congenital microphthalmia, cataracts, and retinal inflammation and develop locomotor deficiencies due to specific axonal degeneration in dorso-lateral tracts of the spinal cord, which relay proprioceptive signals from the hind limbs to the cerebellum. YME1L deficiency is associated with defects in the axonal transport of mitochondria in cultured neurons. While we observe disturbed OPA1 processing and mitochondrial fragmentation in these mice, restoration of tubular mitochondria upon loss of a second mitochondrial protease, OMA1, deteriorates axonal degeneration demonstrating that impaired mitochondrial proteostasis rather than mitochondrial fragmentation causes axonal transport defects and neurological dysfunction.

**Impact**

Our study provides insight into pathogenic mechanisms in a mitochondriopathy associated with YME1L and highlights the importance of mitochondrial proteostasis for eye development and axonal maintenance. We demonstrate for the first time that stress-induced OPA1 processing and mitochondrial fragmentation has highly tissue-specific consequences and unravel a novel pro-survival function of OMA1 in neurons.

## For more information
Genes encoding for proteases associated with human diseases:
AFG3L2 (OMIM: 610246; 614487), CLPP (OMIM: 614129), HTRA2 (OMIM: 610297), IMMP2L (OMIM: 137580), LONP1 (OMIM: 600373), PARL (OMIM: 168600), PMPCB (OMIM: 229300), SPG7 (OMIM: 607259), XPNPEP3 (OMIM: 6131599), YME1L (OMIM: 617302).

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
