## [Review Process File · EMBO Molecular Medicine]

Loss of the mitochondrial *i*-AAA protease YME1L leads to ocular dysfunction and spinal axonopathy

Hans-Georg Sprenger, Gulzar Wani, Annika Hesseling, Tim König, Maria Patron, Thomas MacVicar, Sofia Ahola, Timothy Wai Esther Barth, Elena I. Rugarli, Matteo Bergami and Thomas Langer

Review timeline:

Submission date:	01 May 2018
Editorial Decision:	04 June 2018
Revision received:	21 August 2018
Editorial Decision:	02 October 2018
Revision received:	04 October 2018
Accepted:	08 October 2018

Editor: Céline Carret

Transaction Report:

1st Editorial Decision

04 June 2018

Thank you for the submission of your manuscript to EMBO Molecular Medicine. We have now heard back from the three referees whom we asked to evaluate your manuscript.

You will see from the reports pasted below that overall the study is found interesting and should be further considered. However, more explanations and clarifications are needed, along with additional experiments as suggested to strengthen the findings. Please note that for referee 2, the clinical relevance is unclear while better appreciated for referee 3, nevertheless, this aspect would need to be improved by further discussion, as it remains a strong point for our scope and interest. In addition, the editorial advisor we consulted with prior to sending the paper out for review noted that the recovery of mito-network did not do good for the phenotype, and wondered if the worsening could be due to stalled macro-autophagy because of fused network inhibiting the organellar turnover. We would like to suggest investigating this mechanistic aspect of the work to further improve the data.

We would welcome the submission of a revised version within three months for further consideration and would like to encourage you to address all the criticisms raised as suggested to improve conclusiveness and clarity. Please note that EMBO Molecular Medicine strongly supports a single round of revision and that, as acceptance or rejection of the manuscript will depend on another round of review, your responses should be as complete as possible.

I look forward to receiving your revised manuscript.

***** Reviewer's comments *****

Referee #1 (Comments on Novelty/Model System for Author):

The manuscript of Sprenger et al describes the phenotype of a mouse model with a conditional deletion of the YME1L gene in the nervous system. YME1L is a inter membrane mitochondrial AAA metalo-protease involved in the control of membrane dynamics through the processing of OPA1 and in proteostasis.

The quality of the work and its presentation (manuscript + figures) is excellent, with a rational and convincing construction of the experimental flow.

The technical quality of the work is excellent with impressive amounts of histological, cellular, molecular and biochemical data, emphasizing the consequences of YME1L deletion on the eye, brain and spinal cord physiology of the animals. In addition some in vivo experiments provide clear-cut correlations between the neuron alterations and their consequences on the animal behaviour. Statistics analyses are adequate.

The novelty of the work is important, as to date, the function of YME1L has only be tested in the heart, by the same team using a conditional KO in this organ. In addition, the paper presents the consequences of the isolated OMA1 deletion or combined to YME1L deletion, disclosing an antagonist activity of YME1L and OMA1 on mitochondrial dynamics, while a combined effect on the phenotype neuronal phenotype of the double KO, thus pointing on a critical convergent role of both proteins on proteostasis.

Medical impact is interesting, because one homozygous YME1L mutations has been identified in individuals with a severe mitochondrial syndrome including intellectual disability, motor developmental delay, expressive speech delay, optic nerve atrophy and hearing impairment, as the cardinal features. Nevertheless the phenotype of the mouse presented in this work does not reproduce the one found in humans, most probably because the effect of a deletion is different from the effect of a missense mutation and because the YME1L deletion is restricted to the neuronal system in the mouse model, while ubiquitous in the patients.

The model system is adequate to the objectives of the work: to study the role of YME1L in the nervous system, and the experimental flow is well adapted to these objectives.

Referee #1 (Remarks for Author):

The manuscript of Sprenger et al describes the phenotype of a mouse model with a conditional deletion of the YME1L gene in the nervous system, and raises few questions concerning the results that could be commented in the discussion:

The anterior chamber of the eye ball is affected by YME1L deletion, whereas the posterior chamber and in particular the retina is barely affected. Two other examples of a gene deletion (HCCS and COX7B) encoding mitochondrial proteins are leading to such a phenotype. Is there a cross talk between the function of these 2 genes and that of YME1L?

The retinal size looks somehow increased in the NYKO mouse, with abnormally localized cells in the OPL and eventually in the IPL. Also the NYKO soma size and/or the space between the retinal cells seem larger than in controls (Fig1B). Is there less dendrites or less mitochondria in dendrites and axons in NYKO neurons than in WT, which could explain a lower level of mitochondrial axonal/dendrite transport and eventually a larger number of mitochondria in the soma?

An inflammation process has been found in the NYKO retina with increased levels of GFAP and pro-inflammatory cytokines. Such an inflammation process has recently been reported in an Opa1 muscle KO condition, involving NF-KB, FGF21 and the mtDNA (Rodriguez-Nuevo et al, EMBO J 37 (10), 2018). Possible convergent or divergent pathways leading to these inflammation processes related to mitochondrial dynamic actors could be probed and discussed.

Morphometric evaluation of the brain structures shown in Fig3 could emphasize the absence of differences between the two genotypes.

The resolution of the pictures from Fig5A could be better and eventually emphasize normal or abnormal dendrite arborisation and mitochondrial distribution, as already illustrated in neurons altered for other genes involved in mitochondrial dynamics.

Referee #2 (Remarks for Author):

This is an excellent report on an aspect of mitochondrial metabolism and maintenance, but the extrapolation to the clinical condition is probably the weakest part of the manuscript.

The introduction provides a clear overview of the state of relevant knowledge on fission/fusion and the regulation of mitochondrial morphology under healthy and mouse models mimicking human disease conditions. It sets out what is known about YME1 functions and interactions with Oma1, and the enigmas that remain particularly with respect to neuronal tissue specificity.

Results are clearly described and accurately represent the data provided. The lack of effect in the heterozygous mice is quite striking in figs 1 and 2.

The selective and developmentally regulated expression of inflammatory markers was quite distinctive and perhaps surprising but very clear. The difference between IL6 and TNF/IL1b was very clear.

Fig 8 legend includes text that describes and interprets the data. This would be better in the main text and not the legend.

Not sure why Cre-GFP was expressed and not clear from the fig that this is the case. The phrasing is a little disingenuous "By contrast, in Yme1l-deficient neurons the proportion of moving mitochondria was almost halved". Although this is accurate it makes the difference seem far more dramatic. It does not make the point that is clear in the histogram, that in wt over 60% are stationary cf ~78% in -/- animals. Directionality of mitochondrial movement would have been interesting to include. E.g. were the mitochondria all moving away from the nucleus in the WT but more randomly in the -/- ?

The mitochondria/axon in cultured neurons was decreased but looking at the same parameter in brain/cord slices would have shown whether this was also true in the physiological context.

The data clearly show that the defects resulting from YME1L are exacerbated with age. The change in mitochondrial morphology (17,9% of the mitochondria in the dorso-lateral tract) was striking, but a curious feature was that although a proportion was dramatically affected, the remainder looked perfectly normal. Was this representative or were the EM images displaying the most distorted looking mitos?

If the mitochondrial distortion and inflammation do not initiate the degeneration process, are they a consequence of a different pathway triggered by the loss of YME1L, which isn't fragmentation, since the restoration of the fission does not overcome the neurological problems.

The text gives the impression that the change in mitochondrial length is almost universal ("These experiments identify impaired OPA1 processing and mitochondrial fragmentation"), however neither the images nor the histogram show this with an average length decreasing from ~0.9 to ~0.7um or 1.25 to 1.6um, rather than generating a punctate population. The discussion of the fusion/fission status and mitochondrial function and effect on neuronal cells is a little confusing as it

equates loss of a network with dysfunction and then the restoration of a network as an improvement but not in the neuronal function.

Changes in ability to feed in the -/- mice is indicated as potentially affecting the loss of body weight/mass, this makes the data on this difficult to evaluate.

'THE PAPER EXPLAINED - Does not need a hyphen after "cell type-"
Need to remove 's' from regulates in "YME1L that regulates..."

Legends

"26-week-old" hyphens are not required. This is found in a number of the figure legends.

In statistical methods

"We have not performed any sp....." Spelling needs correcting. I would imagine that statistics on the sup figures should go with the supplemental materials not in the main methods.

Referee #3 (Comments on Novelty/Model System for Author):

The experiments are well done. The mouse models are novel and reflect features of human disease associated with YME1L mutations. The main message is that neurodegeneration proceeds independently of mitochondrial "fragmentation" which is often used inappropriately in the literature as a disease surrogate.

Referee #3 (Remarks for Author):

In this manuscript, Sprenger et al. investigate a novel conditional KO mouse for YME1L (NYKO), which causes a mitochondriopathy in humans when mutated. YME1L KO driven by nestin-Cre produces defects in the eye and degeneration in the spinal cord, but not in other CNS areas. These defects are preceded by mitochondrial fragmentation and L-OPA1 processing but not by an overt respiratory defect until the mouse reaches a terminal disease stage at 30 weeks of age. Using cortical neuronal cultures, the authors also identify mitochondrial motility defects in YME1L KO cells. Importantly, they demonstrate that YME1L KO-induced neurodegeneration is not caused by mitochondrial fragmentation, as simultaneous KO of YME1L and OMA1 (NYKOKO) restores L-OPA1 and tubular mitochondria, but surprisingly accelerates and extends the degenerative phenotype. Based on these observations the authors posit that a loss of proteostatic function in mitochondria, rather than mitochondrial fusion or cristae alterations, is the likely culprit of neurodegeneration in NYKO mice.

This mouse model is an important step toward understanding the function of YME1L in the CNS. There are similarities between the alterations in mice and the human phenotype associated with YME1L recessive mutations. Therefore, overall the paper provides useful phenotypic data that can help understanding the pathology of the human genetic disorder. The finding that mitochondrial morphological changes can be dissociated from degeneration is very intriguing. The manuscript clearly shows the dissociation between mitochondrial tubularity and disease. Demonstration that YME1L KO leads to impaired mitochondrial proteostasis as a mechanism of neurodegeneration would be important. However, a further characterization of this mouse and additional studies *in vitro* could more convincingly support this conclusion.

Specific points

1. The authors propose that there are cell type-specific deficits caused by YME1L KO that affect spinal cortical and spinal cerebellar tracts, but do not affect cortical or cerebellar neurons, among others. This is an important conclusion of the study. However, the authors demonstrate altered L-OPA1 processing and STARD7/PRELID1 accumulation in the forebrain of NYKO mice. This apparent discrepancy needs to be addressed by evaluating different brain regions and spinal cord in parallel. It is possible that the extent of the alterations is different in CNS regions.
2. The reason for increased energy expenditure both during the day and at night is unclear. This could be better discussed.

3. The reason for using cultured cortical neurons to establish alterations in mitochondrial motility as a potential mechanism of degeneration is somehow unclear, as cortical regions are reported to not degenerate in this mouse. Again, it is possible that cortical neurons are not susceptible to damage secondary to these alterations. Nevertheless, it is important to compare the cortical neurons with affected neurons. Perhaps, the DRGs would be more affected, given the proprioceptive neuron degeneration.
4. Data demonstrating the assertion that there is, "no abnormal morphology or degeneration" in cortical neurons should be provided. The data provided in Figure 5A should be expanded to include disease relevant parameters (e.g. axon/dendrite length or branching, quantification of synaptic number and/or a quantification of mitochondria resident at synapses).
5. Muscle weakness in the hind limbs suggests degeneration of the upper or lower motor neurons, but these are reported as unaffected. It is possible that distal portions of the motor axons or the synapses are degenerating.
6. The authors demonstrate processing of L-OPA1 and accumulation of PRELID1 and STARD7 in the spinal cord of YME1L KO mice (Figure S5B) with no discernable respiratory deficit at 6 weeks (Figure 6A). They also present data of the same processes in the forebrain with no cortical degeneration. These data suggest that the degenerative phenotype observed in the affected neurons mice requires additional steps beyond the defect in proteolysis, yet this is not addressed.
7. It is difficult to conclude that cristae abnormalities follow axonal degeneration in the NYKO mouse, because both were absent at 6 weeks, while both detected at 17 weeks.
8. The connection between increased STARD7 or PRELID1 and axonal degeneration is not made clear in this manuscript, nor is it obvious from the literature. If the simultaneous KO of OMA1 exacerbates degeneration caused by YME1L in NYOKO mice by increased proteotoxicity, STARD7 and PRELID1 accumulation should be aggravated. This should be considered to support this conclusion. Alternatively, OMA1 KO may aggravate the disease through a parallel, non-synergetic mechanism.
9. The KYOKO mouse has normal mitochondrial length, but it is unclear whether the cristae alterations are also rescued in this double KO mouse. This is surmised from the L-OPA1 levels and the light microscopy, but not demonstrated by TEM.
10. YME1L is required for degradation of the TIM complex subunit TIM17a as a proteostatic mechanism. So, an analysis of TIM17a would be informative in this case.
11. YME1L KO in the in vitro system should be confirmed.
12. In describing neuronal live imaging, the statement "Morphological discrimination between axonal and other dendritic processes was performed..." should be clarified.
13. In NYKO cortical neurons there are less mitochondria in the axonal area (processes). Does that correspond to increased pool of mitochondria in the soma or is there a generalized depletion of mitochondria?

1st Revision - authors' response

21 August 2018

Reply to editorial advisor:

In addition, the editorial advisor we consulted with prior to sending the paper out for review noted that the recovery of mito-network did not do good for the phenotype, and wondered if the worsening could be due to stalled macro-autophagy because of fused network inhibiting the organellar turnover. We would like to suggest investigating this mechanistic aspect of the work to further improve the data.

To address whether stalled macro-autophagy could be causative for the worsening of the phenotype in NYOKO mice, we analyzed spinal cords isolated from 6-7 week old mice by immunoblotting and

quantitative real-time PCR. The autophagy marker proteins p62 and LC3B did not accumulate at different levels when comparing spinal cord protein lysates from mice depleted for YME1L, OMA1 or both proteases (new Appendix Fig S6D). Moreover, we did not observe differences in the mRNA expression of autophagy related genes when comparing NYOKO mice spinal cords to control tissue (new Appendix Fig S6E). In addition, immunoblot analysis of different mitochondrial marker proteins revealed no effect on overall mitochondrial mass between the different genotypes (new Appendix Fig S6F). Taken together, our experiments suggest that the worsening of the phenotype in NYOKO mice, lacking both YME1L and OMA1, is independent of stalled macro-autophagy.

Reply to Reviewer 1:

Comments on Novelty/Model System:

The manuscript of Sprenger et al describes the phenotype of a mouse model with a conditional deletion of the YME1L gene in the nervous system. YME1L is a inter membrane mitochondrial AAA metalo-protease involved in the control of membrane dynamics through the processing of OPA1 and in proteostasis.

The quality of the work and its presentation (manuscript + figures) is excellent, with a rational and convincing construction of the experimental flow.

The technical quality of the work is excellent with impressive amounts of histological, cellular, molecular and biochemical data, emphasizing the consequences of YME1L deletion on the eye, brain and spinal cord physiology of the animals. In addition some in vivo experiments provide clear-cut correlations between the neuron alterations and their consequences on the animal behaviour. Statistics analyses are adequate.

The novelty of the work is important, as to date, the function of YME1L has only be tested in the heart, by the same team using a conditional KO in this organ. In addition, the paper presents the consequences of the isolated OMA1 deletion or combined to YME1L deletion, disclosing an antagonist activity of YME1L and OMA1 on mitochondrial dynamics, while a combined effect on the phenotype neuronal phenotype of the double KO, thus pointing on a critical convergent role of both proteins on proteostasis.

Medical impact is interesting, because one homozygous YME1L mutations has been identified in individuals with a severe mitochondrial syndrome including intellectual disability, motor developmental delay, expressive speech delay, optic nerve atrophy and hearing impairment, as the cardinal features. Nevertheless the phenotype of the mouse presented in this work does not reproduce the one found in humans, most probably because the effect of a deletion is different from the effect of a missense mutation and because the YME1L deletion is restricted to the neuronal system in the mouse model, while ubiquitous in the patients.

The model system is adequate to the objectives of the work: to study the role of YME1L in the nervous system, and the experimental flow is well adapted to these objectives.

Remarks for Author:

The manuscript of Sprenger et al describes the phenotype of a mouse model with a conditional deletion of the YME1L gene in the nervous system, and raises few questions concerning the results that could be commented in the discussion:

The anterior chamber of the eye ball is affected by YME1L deletion, whereas the posterior chamber and in particular the retina is barely affected. Two other examples of a gene deletion (HCCS and COX7B) encoding mitochondrial proteins are leading to such a phenotype. Is there a cross-talk between the function of these 2 genes and that of YME1L?

The depletion of cytochrome c heme lyase HCCS, an essential player of the respiratory chain, causes increased cell death via caspase-9 activation and recapitulates microphthalmia with linear skin lesions (MLS) in a medaka model (Indrieri et al, 2013). The authors have demonstrated that respiratory chain impairment and overproduction of reactive oxygen species (ROS) trigger caspase-9 activation in this model. Similarly, deletion of the structural subunit of cytochrome c oxidase COX7B causes respiratory dysfunction and MLS (Indrieri et al, 2012). To address whether there is a cross talk between the function of these two genes and that of YME1L, we first analyzed mRNA

expression level of *Hccs* and *Cox7b* in NYKO mice. We observed similar expression levels of both genes in retinas of NYKO and control mice (new Appendix Fig S1C). Moreover, while reduced in the HCCS medaka model, we did not observe any reduction of cytochrome c levels in NYKO mice (new Appendix Fig S1D).

To investigate a potential overlap in downstream signaling pathways, we monitored mRNA expression levels of ROS scavenger enzymes and the activation of caspase-8 and 9 in NYKO mice tissues. The loss of YME1L did not result in increased expression of ROS scavenger proteins (such as SOD1, SOD2, catalase or GPX1) nor in accumulation of the active forms of caspase-8 and 9 (new Appendix Fig S1E and F).

We therefore conclude that YME1L protects against microphthalmia via a pathway independent of respiratory chain dysfunction, ROS overproduction and caspase-9 signaling. Notably, we observed decreased levels of a proteolytic product of caspase 9 in NYKO retinas pointing rather to an inhibition of the pathway. These findings are supported by the absence of a respiratory dysfunction in tissues of NYKO mice before onset of pathology as shown in Figure 6A. We refer to these observations in the discussion of the revised manuscript.

The retinal size looks somehow increased in the NYKO mouse, with abnormally localized cells in the OPL and eventually in the IPL. Also the NYKO soma size and/or the space between the retinal cells seem larger than in controls (Fig1B). Is there less dendrites or less mitochondria in dendrites and axons in NYKO neurons than in WT, which could explain a lower level of mitochondrial axonal/dendrite transport and eventually a larger number of mitochondria in the soma?

We carefully quantified axonal and dendritic lengths in cultured, YME1L-deficient neurons but did not observe significant changes (new Figs. 5B-E and EV3A). Thus, the decreased number of mitochondria in axons is unlikely caused by a decreased number of neurites. Notably, our *in vivo* data argue against a general depletion of mitochondria in the absence of YME1L (new Appendix Fig S6D-F) The swollen appearance of the retina may be attributed to the severe neuroinflammation being present in the tissue.

An inflammation process has been found in the NYKO retina with increased levels of GFAP and pro-inflammatory cytokines. Such an inflammation process has recently been reported in an Opa1 muscle KO condition, involving NF-KB, FGF21 and the mtDNA (Rodriguez-Nuevo et al, EMBO J 37 (10), 2018). Possible convergent or divergent pathways leading to these inflammation processes related to mitochondrial dynamic actors could be probed and discussed.

Loss of OPA1 causes fragmentation of the mitochondrial network similar to what we observe in NYKO mice. However, our results demonstrate that mitochondrial fragmentation is unlikely to be the major cause for inflammation in NYKO mice. Additional deletion of *Oma1* prevents mitochondria to fragment but does not ameliorate neuroinflammation and deteriorates axonal degeneration in NYKO mice. Notably, OPA1 deficient cells are respiratory deficient and accumulate mtDNA at lower levels. Rodriguez-Nuevo and colleagues (2018) could attribute NFκB activation and increased *Fgf21* expression in OPA1-deficient muscles to mtDNA stress and disrupted mitophagy. In contrast, deletion of *Yme1l* did not affect respiration (prior to onset of neurodegeneration) and did not impair the accumulation of mtDNA or general mitophagy (new Appendix FigS1B and S6D-F). Moreover, loss of OPA1 or YME1L affects the expression of NFκB target genes differently. While OPA1 deficiency induces a broad increase in the expression of all tested NF-κB targets (Rodriguez-Nuevo et al, 2018), the response to YME1L depletion affects mainly *Tnfa* and *Asc* (new Fig 1E). We now also analyzed the expression of the cytokine *Fgf21* downstream of the inflammatory process (new Fig 1F). Despite the described different effects on mitochondrial function, both loss of OPA1 in muscle and loss of YME1L in the retina cause increased *Fgf21* expression (new Fig 1F). While our manuscript was in revision, Restelli and colleagues (2018) demonstrated *Fgf21* expression in neurons in several models for neurodegenerative disorders, suggesting that it may be a potent biomarker for neuronal mitochondriopathies in general.

Morphometric evaluation of the brain structures shown in Fig3 could emphasize the absence of differences between the two genotypes.

To emphasize that the overall brain morphology is not affected in NYKO mice, we quantified the thickness of different brain regions and included the data in the manuscript (new Fig 3C).

The resolution of the pictures from Fig5A could be better and eventually emphasize normal or abnormal dendrite arborisation and mitochondrial distribution, as already illustrated in neurons altered for other genes involved in mitochondrial dynamics.

We now show pictures with higher resolution in the new Fig. 5A. To quantify dendrite arborisation, we analyzed neurons at DIV7, to allow for a clear quantification of dendritic/axonal arborisation in individual neurons. Quantification of dendrite number as well as number of axonal branch points revealed no differences in the absence of YME1L (new Figs 5B-E, EV3A). Whereas the total axonal length per cell was also not affected (Fig 5B), the total dendritic length per cell was slightly reduced in *Yme1l*^{-/-} neurons (Fig 5C).

Reply to Reviewer 2:

This is an excellent report on an aspect of mitochondrial metabolism and maintenance, but the extrapolation to the clinical condition is probably the weakest part of the manuscript.

The introduction provides a clear overview of the state of relevant knowledge on fission/fusion and the regulation of mitochondrial morphology under healthy and mouse models mimicking human disease conditions. It sets out what is known about YME1 functions and interactions with Oma1, and the enigmas that remain particularly with respect to neuronal tissue specificity.

Results are clearly described and accurately represent the data provided. The lack of effect in the heterozygous mice is quite striking in figs 1 and 2.

The selective and developmentally regulated expression of inflammatory markers was quite distinctive and perhaps surprising but very clear. The difference between IL6 and TNF/IL1b was very clear.

Fig 8 legend includes text that describes and interprets the data. This would be better in the main text and not the legend.

The text was removed from the legend to Fig. 8 and included in the main text (p. 12).

Not sure why Cre-GFP was expressed and not clear from the fig that this is the case.

In experiments shown in Fig. 5, cortical neurons were isolated from *Yme1l*^{fl/fl} animals and either CAG-GFP or CAG-Cre-IRES-GFP plasmids were expressed to generate control and *Yme1l*^{-/-} neurons, respectively. This approach was chosen to directly compare WT and *Yme1l*^{-/-} neurons obtained from the very same preparation, but also to disclose cell-autonomous effects of YME1L during neuronal development and maturation *in vitro*. Only GFP expressing neurons were quantified. We now also verified deletion of *Yme1l* upon Cre nucleofection (new Appendix Fig S3). The legend to Fig. 5 has been modified to clarify the experimental setup.

The phrasing is a little disingenuous "By contrast, in Yme1l-deficient neurons the proportion of moving mitochondria was almost halved". Although this is accurate it makes the difference seem far more dramatic. It does not make the point that is clear in the histogram, that in wt over 60% are stationary cf ~78% in -/- animals.

We modified the phrasing in that paragraph to better describe the proportions of stationary vs motile mitochondria, and stated the actual percentages corresponding to our quantification.

Directionality of mitochondrial movement would have been interesting to include. E.g. were the mitochondria all moving away from the nucleus in the WT but more randomly in the -/- ?

Our initial experimental setup did not allow to unambiguously distinguish between mitochondria moving in the anterograde or retrograde direction, due to the marked density, branching and intersection of individual axonal fibers at DIV10-14. To comply with the request of the reviewer, we

performed trafficking experiments in neurons at DIV7 when neuronal complexity still allows to distinguish between mitochondria moving in the anterograde or retrograde direction in axons that can be tracked back to the body or up to the growth cone. These experiments revealed that the loss of YME1L impairs mitochondrial movement in the anterograde direction while mitochondria moving in the retrograde direction were not affected (new Figs. 5F and G). These new data are in line with the overall reduction of mitochondrial trafficking detected by DIV10-14 and with the reduced density of mitochondria in the axon (Fig 5J), although the overall mitochondrial mass is not affected *in vivo* (new Appendix Fig S6D-F).

The mitochondria/axon in cultured neurons was decreased but looking at the same parameter in brain/cord slices would have shown whether this was also true in the physiological context.

We have started to determine the mitochondria/axon ratio using TEM images of dorso-lateral tracts of available mice at 6-7 weeks of age and at 31-32 weeks of age. As shown in Fig. R1, we observed a tendency towards decreased mitochondria/axon with age, which however was not statistically significant based on the limited number of animals analyzed. We therefore prefer not to include these data in the manuscript.

Figure R1 - Mitochondria/axon ratio in the dorso-lateral tracts of spinal cords in NYKO mice.

A Quantification of TEM images showing mitochondrial numbers/axon ratios in dorso-lateral tracts at 6-7 weeks of WT (n = 4, 1663 axons and 1209 mitochondria quantified) and NYKO (n = 4, 2770 axons and 1792 mitochondria quantified) mice. **B** Quantification of TEM images showing mitochondrial numbers/axon in dorso-lateral tracts at 31-32 weeks of WT (n = 3, 1451 axons and 727 mitochondria quantified) and NYKO (n = 3, 1854 axons and 742 mitochondria quantified) mice. Unpaired t-test, ns = not significant. Data are means \pm SEM.

The data clearly show that the defects resulting from YME1L are exacerbated with age. The change in mitochondrial morphology (17,9% of the mitochondria in the dorso-lateral tract) was striking, but a curious feature was that although a proportion were dramatically affected, the remainder looked perfectly normal. Was this representative or were the EM images displaying the most distorted looking mitos ?

The TEM images shown in Fig. 6 are representative for the morphology of the mitochondrial population in the dorso-lateral tracts of NYKO mice. We have included a careful quantification of mitochondrial morphology to emphasize that only a fraction of mitochondria shows an altered morphology. Notably, NYKO mitochondria were indistinguishable for wild type mitochondria before onset of the pathology, suggesting that a disturbed ultrastructure of mitochondria is not causative but rather a secondary event.

If the mitochondrial distortion and inflammation do not initiate the degeneration process, are they a consequence of a different pathway triggered by the loss of YME1L, which isn't fragmentation, since the restoration of the fission does not overcome the neurological problems.

We observed respiratory deficiencies and disturbances in mitochondrial ultrastructure only after onset of axonal degeneration, indicating that these deficiencies do not initiate the degeneration process. As we also excluded mitochondrial fragmentation as being causative and in view of the multiple proteolytic substrates of YME1L (accumulating in YME1L deficient neurons), we indeed propose that defective mitochondrial proteostasis affecting a different pathway results in axonal degeneration. We have modified the discussion to clarify this point.

The text gives the impression that the change in mitochondrial length is almost universal ("These experiments identify impaired OPA1 processing and mitochondrial fragmentation"), however neither the images nor the histogram show this with an average length decreasing from ~0.9 to ~0.7µm or 1.25 to 1.6µm, rather than generating a punctate population. The discussion of the fusion/fission status and mitochondrial function and effect on neuronal cells is a little confusing as it equates loss of a network with dysfunction and then the restoration of a network as an improvement but not in the neuronal function.

We agree with the reviewer that mitochondrial fragmentation is in general a rather unprecise term to describe an altered morphology of the mitochondrial network characterized by shorter tubules. In fact, multiple authors describe fragmented mitochondrial networks in the literature but refer to morphologies characterized by significantly different tubule length. This is in fact one reason why we show histograms (rather than an average tubule length) that result from a careful quantification of mitochondrial tubule length in the population. Moreover, heterogeneity within a tissue can add an additional level of complexity. Histograms show the mitochondrial length distributions in specific areas of the tissue, which may include both cells deleted for *Yme1l* and cells still harboring YME1L. Together, we feel that histograms are the most appropriate possibility to illustrate the alterations in mitochondrial morphology in the absence of YME1L and reconsidered our wording throughout the manuscript.

Changes in ability to feed in the -/- mice is indicated as potentially affecting the loss of body weight/mass, this makes the data on this difficult to evaluate.

The changes in ability to feed NYKO mice were specifically noted during PhenoMaster experiments where food containers hung at the sensor and were more difficult to access than in normal cages. To exclude that the accessibility of food affects the data acquisition, we provided food from the bottom of the cage. Therefore, the increased energy expenditure monitored of NYKO mice could explain the reduction in body weight and fat mass. However, since no determination of actual food intake was performed it is possible that deletion of *Yme1l* in the nervous system additionally affects feeding behavior via a direct effect on hunger and satiety controlling neurons or rather indirect mechanism due to the pathology of these mice.

THE PAPER EXPLAINED - Does not need a hyphen after "cell type-"

Need to remove 's' from regulates in "YME1L that regulates..."

Legends

"26-week-old" hyphens are not required. This is found in a number of the figure legends.

In statistical methods

"We have not performed any sp....." Spelling needs correcting. I would imagine that statistics on the sup figures should go with the supplemental materials not in the main methods.

We introduced these changes in the revised manuscript as suggested.

Reply to Reviewer 3:

Comments on Novelty/Model System:

The experiments are well done. The mouse models are novel and reflect features of human disease associated with YME1L mutations. The main message is that neurodegeneration proceeds independently of mitochondrial "fragmentation" which is often used inappropriately in the literature

as a disease surrogate.

Remarks for Author:

In this manuscript, Sprenger et al. investigate a novel conditional KO mouse for YME1L (NYKO), which causes a mitochondriopathy in humans when mutated. YME1L KO driven by nestin-Cre produces defects in the eye and degeneration in the spinal cord, but not in other CNS areas. These defects are preceded by mitochondrial fragmentation and L-OPA1 processing but not by an overt respiratory defect until the mouse reaches a terminal disease stage at 30 weeks of age. Using cortical neuronal cultures, the authors also identify mitochondrial motility defects in YME1L KO cells. Importantly, they demonstrate that YME1L KO-induced neurodegeneration is not caused by mitochondrial fragmentation, as simultaneous KO of YME1L and OMA1 (NYOKO) restores L-OPA1 and tubular mitochondria, but surprisingly accelerates and extends the degenerative phenotype. Based on these observations the authors posit that a loss of proteostatic function in mitochondria, rather than mitochondrial fusion or cristae alterations, is the likely culprit of neurodegeneration in NYKO mice.

This mouse model is an important step toward understanding the function of YME1L in the CNS. There are similarities between the alterations in mice and the human phenotype associated with YME1L recessive mutations. Therefore, overall the paper provides useful phenotypic data that can help understanding the pathology of the human genetic disorder. The finding that mitochondrial morphological changes can be dissociated from degeneration is very intriguing. The manuscript clearly shows the dissociation between mitochondrial tubularity and disease. Demonstration that YME1L KO leads to impaired mitochondrial proteostasis as a mechanism of neurodegeneration would be important. However, a further characterization of this mouse and additional studies in vitro could more convincingly support this conclusion.

Specific points

1. The authors propose that there are cell type-specific deficits caused by YME1L KO that affect spinal cortical and spinal cerebellar tracts, but do not affect cortical or cerebellar neurons, among others. This is an important conclusion of the study. However, the authors demonstrate altered L-OPA1 processing and STARD7/PRELID1 accumulation in the forebrain of NYKO mice. This apparent discrepancy needs to be addressed by evaluating different brain regions and spinal cord in parallel. It is possible that the extent of the alterations is different in CNS regions.

To evaluate the relative accumulation of YME1L substrate proteins in different brain regions and spinal cord in parallel, we isolated proteins from different CNS regions from 31-32 week old animals and analyzed the samples by SDS-PAGE and immunoblotting of the same membrane (new Fig 7E). We did not observe differences in mitochondrial mass/protein amount loaded between the tissues, allowing a direct comparison of protein levels of YME1L substrates. YME1L substrates indeed accumulated at different levels in different regions of the CNS of NYKO mice (new Fig 7E), substantiating our hypothesis that the loss of YME1L has differential effects on mitochondrial proteostasis in various CNS regions. It should be noted, however, that the accumulation of the specific substrates tested did not correlate with the appearance of tissue pathology, which at this age is restricted to the spinal cord.

We conclude that YME1L regulates mitochondrial protein levels in different regions of the CNS (consistent with its ubiquitous expression). However, YME1L-dependent maintenance of mitochondrial proteostasis is particularly important in neurons located in the dorso-lateral tracts of the spinal cord. It is conceivable that these neurons are more susceptible to disturbances in YME1L-dependent pathways (such as intramitochondrial lipid trafficking or mitochondrial protein import) or to disturbances in YME1L-mediated protein quality control. Alternatively, differences in the neuronal vulnerability may result from cell-specific compensatory mechanisms.

2. The reason for increased energy expenditure both during the day and at night is unclear. This could be better discussed.

Differences in food intake are unlikely to cause the observed differences in energy expenditure. It is possible that deletion of *Yme1l* in the nervous system affects feeding behavior via a direct effect on hunger and satiety controlling neurons. Alternatively, the increased energy expenditure may be an

indirect consequence of the neuronal pathology of the mice. We have extended the relevant parts in the discussion.

3. The reason for using cultured cortical neurons to establish alterations in mitochondrial motility as a potential mechanism of degeneration is somehow unclear, as cortical regions are reported to not degenerate in this mouse. Again, it is possible that cortical neurons are not susceptible to damage secondary to these alterations. Nevertheless, it is important to compare the cortical neurons with affected neurons. Perhaps, the DRGs would be more affected, given the proprioceptive neuron degeneration.

The analysis of cortical neurons which do not degenerate in the absence of YME1L allowed us to exclude secondary effects on mitochondrial motility, where mitochondrial trafficking could be affected due to axonal injury rather than the loss of YME1L directly (Mar et al, 2014). The discovery that YME1L is required for efficient axonal transport of mitochondria in otherwise healthy neurons (Fig. 5) demonstrates that the protease controls mitochondrial transport independent of neuronal injury and identifies YME1L as a novel factor being involved in mitochondrial motility. Moreover, the experiments from our laboratory together with previously published work by other groups indeed suggest that neurons located in the spinal cord are more sensitive to defects in mitochondrial trafficking than cortical neurons, which are affected at much later stages (Lopez-Domenech et al, 2016; Nguyen et al, 2014).

4. Data demonstrating the assertion that there is, "no abnormal morphology or degeneration" in cortical neurons should be provided. The data provided in Figure 5A should be expanded to include disease relevant parameters (e.g. axon/dendrite length or branching, quantification of synaptic number and/or a quantification of mitochondria resident at synapses).

We have now performed a careful quantification of axon/dendritic length or branching (new Fig 5B-E and EV3A). Whereas we observed a moderate reduction in the length of dendrites, other parameters were not altered (new Fig 5C). Similarly, we did not observe visible alterations in the overall extent of synaptic markers (new Fig EV3B and C).

5. Muscle weakness in the hind limbs suggests degeneration of the upper or lower motor neurons, but these are reported as unaffected. It is possible that distal portions of the motor axons or the synapses are degenerating.

We discuss this possibility now in the revised manuscript.

6. The authors demonstrate processing of L-OPA1 and accumulation of PRELID1 and STARD7 in the spinal cord of YME1L KO mice (Figure S5B) with no discernable respiratory deficit at 6 weeks (Figure 6A). They also present data of the same processes in the forebrain with no cortical degeneration. These data suggest that the degenerative phenotype observed in the affected neurons mice requires additional steps beyond the defect in proteolysis, yet this is not addressed.

As outlined in response to point 1, the accumulation of the specific substrates tested did not correlate with the appearance of tissue pathology. In view of the multiple substrates of YME1L we propose that, rather the accumulation of individual substrates, disturbances in mitochondrial proteostasis impairs the functional integrity of mitochondria and cause the degenerative phenotype. However, we would like to point out that no evidence exists for a non-proteolytic function of YME1L, strongly arguing that the observed phenotypes result from impaired proteolysis in NYKO mice.

7. It is difficult to conclude that cristae abnormalities follow axonal degeneration in the NYKO mouse, because both were absent at 6 weeks, while both detected at 17 weeks.

We agree with the reviewer on this point. However, we would like to emphasize that mitochondrial fragmentation and inflammatory responses but not cristae abnormalities are the first phenotypes detected in NYKO mice. Moreover, while we observe severe axonal degeneration in NYKO mice spinal cords already at 6-7 weeks of age (Fig 8E and F), we now demonstrate that cristae structure is normal at this timepoint in NYKO mice, although degeneration is already apparent (new Appendix FigS6A and B). We therefore favor the possibility that cristae abnormalities do not initiate the degenerative process.

8. *The connection between increased STARD7 or PRELID1 and axonal degeneration is not made clear in this manuscript, nor is it obvious from the literature. If the simultaneous KO of OMA1 exacerbates degeneration caused by YME1L in NYOKO mice by increased proteotoxicity, STARD7 and PRELID1 accumulation should be aggravated. This should be considered to support this conclusion. Alternatively, OMA1 KO may aggravate the disease through a parallel, non-synergetic mechanism.*

YME1L is a pleiotropic protease which is known to regulate the abundance of lipid transfer proteins (PRELID1, STARD7) and the protein translocase subunit TIMM17A, to control mitochondrial morphology via processing of OPA1 and to serve quality control functions by degrading misfolded mitochondrial proteins. It is very likely that additional substrates of YME1L exist which thus has a broad effect on mitochondrial proteostasis. Here, we use STARD7, PRELID1 and TIMM17A as selected marker proteins to monitor the effect of YME1L loss of mitochondrial proteostasis. As outlined under point 1, we do not attribute the observed neurological phenotypes to the accumulation of PRELID1, STARD7 or other individual YME1L substrates but rather propose that impaired mitochondrial proteostasis affects the functional integrity of mitochondria, impairing axonal trafficking and triggering axonal degeneration.

We generated NYOKO mice to examine the role of mitochondrial dynamics for axonal degeneration. The unexpected observation that neurological phenotypes were exacerbated rather than suppressed allowed us to conclude that axonal degeneration is not initiated by mitochondrial fragmentation (which was suppressed in NYOKO mice). We can only speculate why the loss of OMA1 exacerbates the phenotype of NYKO mice, as only a very limited number of OMA1 substrates are presently known. It is conceivable that OMA1 has overlapping proteolytic functions overlapping with YME1L and thus further contributes to disturbances in mitochondrial proteostasis. However, it is equally possible that OMA1 acts on a parallel, non-synergetic pathway.

9. *The NYOKO mouse has normal mitochondrial length, but it is unclear whether the cristae alterations are also rescued in this double KO mouse. This is surmised from the L-OPA1 levels and the light microscopy, but not demonstrated by TEM.*

To comply with the request of the reviewer, we have now analyzed the mitochondrial ultrastructure in dorso-lateral tracts of spinal cords of 6-7 week old NYOKO mice (new Appendix Figs S6A and B). TEM analysis revealed that cristae alterations were not significantly different in NYOKO mice at 6-7 weeks of age when compared to control mice (Fig 6 C and D, Appendix Fig S6A and B). This is consistent with the observed stabilization of L-OPA1 (Fig. 8A).

10. *YME1L is required for degradation of the TIM complex subunit TIM17a as a proteostatic mechanism. So, an analysis of TIM17a would be informative in this case.*

We have monitored TIMM17A levels by SDS-PAGE and immunoblotting in different CNS tissues of 31-32 week old NYKO mice (new Fig 7E) as well as in the spinal cords of 6-7 week old NYKO, NOKO and NYOKO mice (new Fig EV4A). As expected TIMM17A dramatically accumulated in the absence of YME1L but was not affected by the loss of OMA1.

11. *YME1L KO in the in vitro system should be confirmed.*

To confirm deletion of *Yme1l* in the *in vitro* system, we isolated genomic DNA from *Yme1l^{fl/fl}* neuronal cultures transfected with GFP (WT, CAG-GFP) or Cre-GFP (*Yme1l^{-/-}*, CAG-Cre-IRES-GFP) expressing plasmids, respectively, and performed PCR analysis. Amplification of recombinant *Yme1l* alleles (deletion PCR) confirmed deletion of *Yme1l* in our *in vitro* system (Appendix Fig S3). Amplification of *Yme1l* alleles containing loxP sites (loxP PCR) was used to control for the purity of our cultures (Appendix Fig S3). To avoid the analysis of untransfected cells present in the cultures, we selected only GFP positive cells for microscopy.

12. *In describing neuronal live imaging, the statement "Morphological discrimination between axonal and other dendritic processes was performed..." should be clarified.*

We have now better clarified this statement in the methods section.

“Identification of axonal (versus dendritic) mitochondria in live-imaging experiments was achieved by first assessing neuronal morphology in GFP and Cre-GFP expressing neurons. GFP signal (labeling the whole morphology of analyzed neurons) was first acquired to unambiguously identify axonal fibres. According to the age of neurons, axonal identity was defined as (i) processes emerging from the body and extending in length three times or longer than any other primary process, (ii) presence of *en-passant* boutons and/or growth cones at their terminals and (iii) absence of dendritic protrusions. To facilitate the analysis of axonal mitochondria in older cultures (DIV 10-14), fields of acquisitions were chosen to be distant from any cell bodies and dendrites, in areas specifically enriched by axonal fibres visibly decorated by boutons”

13. In NYKO cortical neurons there are less mitochondria in the axonal area (processes). Does that correspond to increased pool of mitochondria in the soma or is there a generalized depletion of mitochondria?

Immunoblot analysis for mitochondrial proteins in NYKO mice or the determination of mtDNA levels did not provide any evidence for a depletion of mitochondria (new Appendix Fig S6F and S1B). However, an accurate determination of the mitochondria in the soma turned out to be technical challenging due to the high density of mitochondria in that area at the stage of analysis.

REFERENCES

- Indrieri A, Conte I, Chesi G, Romano A, Quartararo J, Tate R, Ghezzi D, Zeviani M, Goffrini P, Ferrero I et al (2013) The impairment of HCCS leads to MLS syndrome by activating a non-canonical cell death pathway in the brain and eyes. *EMBO Mol Med* 5: 280-293
- Indrieri A, van Rahden VA, Tiranti V, Morleo M, Iaconis D, Tammara R, D'Amato I, Conte I, Maystadt I, Demuth S et al (2012) Mutations in COX7B cause microphthalmia with linear skin lesions, an unconventional mitochondrial disease. *Am J Hum Genet* 91: 942-949
- Kondadi AK, Wang S, Montagner S, Kladt N, Korwitz A, Martinelli P, Herholz D, Baker MJ, Schauss AC, Langer T et al (2014) Loss of the m-AAA protease subunit AFG(3)L(2) causes mitochondrial transport defects and tau hyperphosphorylation. *EMBO J* 33: 1011-1026
- Lopez-Domenech G, Higgs NF, Vaccaro V, Ros H, Arancibia-Carcamo IL, MacAskill AF, Kittler JT (2016) Loss of Dendritic Complexity Precedes Neurodegeneration in a Mouse Model with Disrupted Mitochondrial Distribution in Mature Dendrites. *Cell Rep* 17: 317-327
- Mar FM, Simoes AR, Leite S, Morgado MM, Santos TE, Rodrigo IS, Teixeira CA, Misgeld T, Sousa MM (2014) CNS axons globally increase axonal transport after peripheral conditioning. *J Neurosci* 34: 5965-5970
- Nguyen TT, Oh SS, Weaver D, Lewandowska A, Maxfield D, Schuler MH, Smith NK, Macfarlane J, Saunders G, Palmer CA et al (2014) Loss of Miro1-directed mitochondrial movement results in a novel murine model for neuron disease. *Proc Natl Acad Sci U S A* 111: E3631-3640
- Restelli LM, Oettinghaus B, Halliday M, Agca C, Licci M, Sironi L, Savoia C, Hench J, Tolnay M, Neutzner A et al (2018) Neuronal Mitochondrial Dysfunction Activates the Integrated Stress Response to Induce Fibroblast Growth Factor 21. *Cell Rep* 24: 1407-1414
- Rodriguez-Nuevo A, Diaz-Ramos A, Noguera E, Diaz-Saez F, Duran X, Munoz JP, Romero M, Plana N, Sebastian D, Tezze C et al (2018) Mitochondrial DNA and TLR9 drive muscle inflammation upon Opa1 deficiency. *EMBO J* 37

Thank you for the submission of your revised manuscript to EMBO Molecular Medicine. We have now received the enclosed reports from the referees that were asked to re-assess it. As you will see the reviewers are supportive and I am pleased to inform you that we will be able to accept your manuscript pending minor editorial amendments.

***** Reviewer's comments *****

Referee #2 (Remarks for Author):

The authors have made extensive additions to the manuscript not just to improve the quality and clarity of the text, but they have also extended the range of experimental approaches and included new data.

The responses to the referees' initial comments are extensive and their points made clearly and rationally.

With respects to the points made by this referee, all the points have been adequately addressed.

Referee #3 (Remarks for Author):

The authors have responded to all my previous concerns by performing new experiments and analyses. The data is very strong and the conclusions convincing.

Corresponding Author Name: Thomas Langer
Journal Submitted to: EMBO Molecular Medicine
Manuscript Number: EMM-2018-09288